



# New insights into the drainage of inundated Arctic polygonal tundra using fundamental hydrologic principles

Dylan R Harp[1], Vitaly Zlotnik[2], Charles J Abolt[1], Brent D Newman[1], Adam L Atchley[1], Elchin Jafarov[1], and Cathy J Wilson[1]

[1]Earth and Environmental Sciences Division, Los Alamos National Laboratory, Los Alamos, NM, 87544
[2]Earth and Atmospheric Sciences Department, University of Nebraska, Lincoln, NE, 68588-0340

**Correspondence:** Dylan Harp (dharp@lanl.gov)

**Abstract.** The pathways and timing of drainage from inundated ice-wedge polygon centers in a warming climate have important implications for carbon flushing, advective heat transport, and transitions from carbon dioxide to methane dominated emissions. This research helps to understand this process by providing the first in-depth analysis of drainage from a single polygon based on fundamental hydrogeological principles. We use a recently developed analytical solution to evaluate the

effects of polygon aspect ratios (radius to thawed depth) and hydraulic conductivity anisotropy (horizontal to vertical hydraulic conductivity) on drainage pathways and temporal depletion of ponded water heights of inundated ice-wedge polygon centers. By varying the polygon aspect ratio, we evaluate the effect of polygon size (width), inter-annual increases in active layer thickness, and seasonal increases in thaw depth on drainage. One of the primary insights from the model is that most inundated ice-wedge polygon drainage occurs along an annular region of the polygon center near the rims. This implies that inundated

polygons are most intensely flushed by drainage in an annular region along their horizontal periphery, with implications for transport of nutrients (such as dissolved organic carbon) and advection of heat towards ice wedge tops. The model indicates that polygons with large aspect ratios and high anisotropy will have the most distributed drainage. Polygons with large aspect ratio and low anisotropy will have their drainage most focused near the their periphery and will drain most slowly. Polygons with small aspect ratio and high anisotropy will drain most quickly.

## 1   Introduction

Polygonal tundra occurs in continuous permafrost landscapes lacking exposed bedrock or active sedimentation (Brown et al., 1997; MacKay, 2000) and is estimated to occupy around 250,000 km$^2$, or around 3% of the Arctic landmass predominantly along the north coasts of Alaska and Siberia (Minke et al., 2007). It is rich in organic carbon (Tarnocai et al., 2009; Hugelius

et al., 2014) which is poised to release to the atmosphere as carbon dioxide or methane under a warming climate causing a potential feedback to climate change (Schuur et al., 2008). The microtopography of polygonal tundra collects precipitation and snow melt events resulting in a slow release from the landscape through the thawed subsurface to surface water drainage





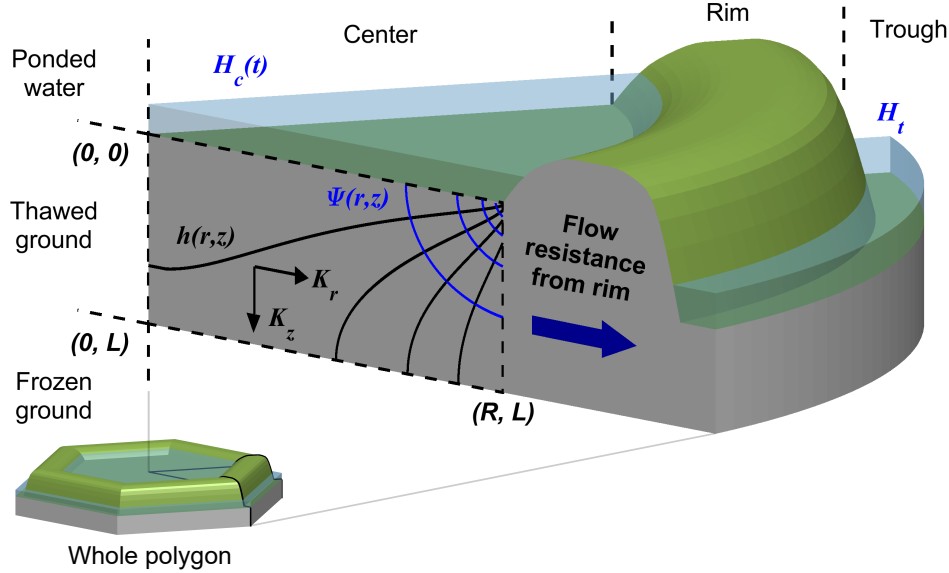

**Figure 1.** Pie-wedge schematic diagram of 3D-axisymmetric analytical model of inundated low-centered polygon drainage. The diagram represents an idealized *pie wedge* slice of a low-centered polygon including a wedge of the ponded center, rim, and trough. Equipotential hydraulic head lines are denoted as $h(r,z)$ and stream lines as $\Psi(r,z)$. $L$ is the depth of the thawed soil layer and $R$ is the radius of the polygon center. The ponded water height ($H_c(t)$) and trough water level ($H_t$) are noted.

networks. This process has important implications for the release of dissolved organic matter from polygon tundra landscapes flushed from the subsurface to surface waters and transport of heat through the subsurface. The role that polygonal tundra

regions play in global terrestrial biogeochemical feedbacks as the Arctic warms necessitates understanding their hydrologic flow regimes, including carbon fluxes from the landscape.

     Polygonal tundra forms in cold environments by the cyclic process of vertical cracking of frozen ground due to thermal contraction, water infiltration into these cracks, freezing of this infiltrated water, and subsequent re-cracking. Over many cycles, this process leads to the growth of subsurface ice wedges connected in polygonal patterns known as ice-wedge polygons

(Liljedahl et al., 2016). Ice-wedge polygons vary in size from several meters to a few tens of meters in diameter and develop over time frames of hundreds to thousands of years (Leffingwell, 1915; Lachenbruch, 1962; MacKay, 2000; Abolt et al., 2018). Thermal expansion of the ice and soil during summer months often results in warping the soil strata forming parallel rims on both sides of the ice wedge and a trough directly above the ice wedge (refer to Figure 1). The peat and mineral soil layering often approximately follow the surface topography, resulting in raised mineral soil under the rims which can impede drainage due

to the mineral soils relatively lower hydraulic conductivity (Hinzman et al., 1991). Ice-wedge polygons with well-formed rims and troughs with a distinct, central topographic depression are referred to as low-centered polygons. Ponding occurs frequently in the centers because the rims can trap water during snow melt or precipitation events. This ponded water subsequently drains through the subsurface to the troughs (Helbig et al., 2013; Koch et al., 2018; Wales et al., 2020). As observed by Helbig et al.



(2013), a high-conductivity, low-capillarity peat layer, which follows the topographic terrain, overlying a lower conductivity
mineral soil, creates a groundwater divide that redistributes water from rims to troughs and centers. After this redistribution,
the drainage is through the underlying mineral soil layer.

Recent observational studies have shaped our current conceptualization of low-centered polygonal tundra hydrology (Boike
et al., 2008; Helbig et al., 2013; Koch et al., 2014; Liljedahl et al., 2016; Koch, 2016; Koch et al., 2018) indicating a system of
inundated polygonal centers surrounded by elevated rims which prevent surface water from flowing overland into surrounding
trough ponds. This land formation results in standing water over the polygon centers, reduces immediate landscape runoff
from precipitation events and increases evaporation (Liljedahl et al., 2016). Additionally, the elevated ponded water in polygon
centers produces hydrological gradients that result in sustained outward flow through the subsurface under the rims, a process
that has been observed at multiple field sites (Helbig et al., 2013; Liljedahl and Wilson, 2016; Koch, 2016; Wales et al.,
2020). This lateral flow controls landscape redistribution of water during the summer months (Helbig et al., 2013) and governs
ponded water budgets (Koch et al., 2014; Koch, 2016) and makes up a notable portion of regional river discharge (King et al.,
2020). The manner and timing with which polygonal tundra landscapes transition from inundated to drained has important
implications for (1) transitions from atmospheric emissions of methane to carbon dioxide (Conway and Steele, 1989; Moore
and Dalva, 1997; Zona et al., 2011; Zhu et al., 2013; O'Shea et al., 2014; Throckmorton et al., 2015; Wainwright et al., 2015;
Lara et al., 2015; Vaughn et al., 2016), (2) dissolved organic carbon emissions to surface waters (Zona et al., 2011; Abnizova
et al., 2012; Laurion and Mladenov, 2013; Larouche et al., 2015; Plaza et al., 2019), (3) biological succession (Billings and
Peterson, 1980; Jorgenson et al., 2015; Wolter et al., 2016), and (4) ground surface deformation (Mackay, 1990; MacKay,
2000; Raynolds et al., 2014; Oehme, 2019).

The microtopographic features of polygonal tundra result in sharply contrasting spatial thermal and hydrologic conditions
(Boike et al., 2008; Zona et al., 2011; Helbig et al., 2013) leading to sharply contrasting spatial biogeochemical conditions
(Newman et al., 2015; Norby et al., 2019). Standing water in polygonal centers show distinctly different chemistry compared
to surface water in troughs and ponds (Zona et al., 2011; Koch et al., 2014, 2018) which has been attributed to shallow
subsurface flow from centers to troughs (Koch, 2016). For example, Koch et al. (2014) observed high nutrient concentrations
in troughs adjacent to centers with low concentrations, indicating inundated center drainage to troughs. The importance of
representing carbon respiration response to changing hydrologic conditions in arctic landscape has motivated the development
of complex biogeochemical modules in Earth system models that are driven by hydrologic conditions (Grant et al., 2017; Bisht
et al., 2018). Nevertheless, A better representation of the tight coupling between hydrologic conditions and biogeochemical
response in Earth system models (Wang et al., 2019) requires well characterized hydrology that dictates transport mechanisms.
Given that the ice-wedge polygon can be considered the fundamental hydrologic landscape unit that initiates landscape scale
surface/subsurface flow and discharge, understanding the characteristic drainage pathways and residence times then provides a
basis to quantify the transport of terrestrially sourced dissolved organic carbon to surface waters, which then can be mineralized
are released back to the atmosphere (Raymond et al., 2013).

While fundamental hydrology dictates that ice-wedge polygon geometry and heterogeneity will explicitly govern subsurface
drainage pathways and time spans, to date, a fundamental hydrological investigation of this has process has not been presented.





The geometric shape of low-centered polygons along with soil hydraulic properties (for example, hydraulic conductivity)
affect the distribution of hydraulic heads that control the pathways and timing of inundated ice-wedge polygon drainage. In
this paper, we use fundamental hydrogeological principles (Harr, 1962; Cedergren, 1968; Freeze and Cherry, 1979; Bear,
1979) to understand effects of geometry and hydraulic properties on inundated ice-wedge polygon drainage. We investigate the
pathways and timing of inundated polygon drainage using a 3D-axisymmetric analytical solution of groundwater heads. This
approach idealizes the thawed subsurface of an inundated low-centered polygon as a thin cylinder overlain with an initial height
of ponded water that drains through the cylindrical thawed soil layer to the surrounding trough (outer vertical boundary; annular
ring defined by a line from $(R, 0)$ to $(R, L)$ in Figure 1). The depth $(L)$ and radius $(R)$ of the thawed soil layer of the polygon
center and horizontal (radial) and vertical hydraulic conductivity are variable within the solution. It is important to note that
we are focusing on drainage of inundated low-centered polygons here, and that biogeochemical investigations suggest that the
drainage of non-inundated low-centered, transition, and high-centered polygons may have different characteristics (Heikoop
et al., 2015; Newman et al., 2015; Wainwright et al., 2015; Wales et al., 2020).

This model provides fundamental insights into the pathways and timing of inundated ice-wedge polygon drainage for poly-
gons with different geometries and degrees of anisotropy. Varying the geometry of the cylinder not only allows us to capture
differences in drainage between different sized polygons (in other words, polygons with different radii), but also seasonal and
inter-annual variations in polygon thaw-layer depth (in other words, as the thawed soil layer increases during the summer or
as the active layer increases from year to year). By allowing for different hydraulic conductivities between the horizontal and
vertical directions, we can evaluate the effects of preferential flow directions on drainage. In addition to geologic layering,
processes such as frost heave and horizontal ice lens thaw can result in preferential horizontal flow in cold climates (Mackay,
1981; Matsuoka and Moriwaki, 1992; Wales et al., 2020). It has also been observed that high conductivity peat layers overlying
lower conductivity mineral soil results in horizontal watershed drainage (McDonnell et al., 1991; Brown et al., 1999; Quinton
and Marsh, 1999). Helbig et al. (2013) found that in polygon tundra, the peat layer quickly redistributed precipitation events
across the subsurface topography from the rims to the centers and troughs. Vertical cracks, voids left from decayed roots, and
animal burrows can result in preferential vertical flow. While the cylindrical geometry and anisotropy considered here do not
cover the potential variations that are known to be present in ice-wedge polygons (for example, anomalous heterogeneities),
the impact of those variations will cause deviations around the base case scenarios considered here.

Although there have been numerous field observations of inundated low-centered polygon drainage to troughs (Helbig et al.,
2013; Koch et al., 2014; Wales et al., 2020), the research here is, to our knowledge, the first in-depth investigation into the
pathways and timing associated with inundated ice-wedge polygon drainage based on fundamental hydrologic principles. Aside
from the field tracer experiments of Wales et al. (2020), previous research has primarily focused on 1D vertical hydrothermal
effects within ice-wedge polygons (Atchley et al., 2015; Harp et al., 2015; Atchley et al., 2016) or ice-wedge surface hydrology
(Boike et al., 2008; Jan et al., 2018a, b). Other researchers have investigated the hydrology of multiple polygons without
investigating the drainage pathways within ice-wedge polygons themselves (Nitzbon et al., 2019). Our analysis provides a new
perspective on inundated polygon hydrogeology indicating the implications of polygon geometry and hydraulic conductivity
anisotropy on drainage pathways and timing.



## 2 Methods

### 2.1 Model parameterization

We selected aspect ratio and anisotropy scenarios based on existing literature and observations. While a pan-Arctic survey of ice-wedge polygon diameters does not currently exist to our knowledge, researchers have provided general characteristics based on extensive observations. Leffingwell (1915) states that ice-wedge polygons have an estimated average diameter of around 15 m. Liljedahl et al. (2016) indicate that low-centered polygons (including rims) can have diameters from 5-30 m. Abolt et al. (2018) state that polygonal formations in ice-wedge tundra are 10-30 m in diameter. Based on digital elevation models, Abolt et al. (2019) calculate that the mean polygon diameter near Utqiaġvik, Alaska is around 15 m with 5th and 95th percentiles of approximately 8 and 33 m, respectively. Using a similar analysis on data from near Prudhoe Bay (Abolt and Young, 2020), the mean polygon diameter is around 13 m with 5th and 95th percentiles of approximately 7 and 30 m, respectively.

Maximum thaw depths are increasing throughout the Arctic (Jorgenson et al., 2006). Deeper thaw depths may ultimately result in low-centered polygon transition to high-centered polygons, at which time center inundation will cease to occur due to rim collapse. Therefore, there is a limit to the thaw depth of interest. Lewkowicz (1994) reported that active layer thickness varied from 40 to 90 cm within the polygonal tundra of Fosheim Peninsula, Ellesmere Island, Canada in 1994. Shiklomanov et al. (2010) reported that polygonal tundra at several study sites near Utqiaġvik, Alaska monitored from 1995 to 2009 had active layer thicknesses from 17 to 47 cm. Based on extensive ground penetrating radar surveys near Utqiaġvik, Alaska in 2013, Jafarov et al. (2017) document that average active layer thickness was 41 cm with a standard deviation of 9 cm. Even for polygons with substantial active layer thickness, it is important to understand the change in drainage throughout the season by evaluating early season thaw depths, which can be captured by larger aspect ratios.

Based on these considerations, we evaluate aspect ratios from 2.5 to 20. For example, given a thawed depth of 1 m, an aspect ratio of 2.5 would represent a 2.5 m radius polygon center, while an aspect ratio of 20 would represent a 20 m radius polygon center, more than covering the observed range. To consider cases with thinner thawed soil layers, larger aspect ratios can be used. For example, given a thawed depth of 0.5 m, an aspect ratio of 20 would represent a polygon with a 10 m radius center. While our selected aspect ratios and anisotropies may not cover all possible scenarios, they do cover a broad enough range of scenarios to draw insights into their effect on inundated ice-wedge polygon drainage.

Comprehensive measurements of hydraulic conductivity anisotropy are lacking from ice-wedge polygons. Based on tracer arrival times, Wales et al. (2020) estimated horizontal conductivities of approximately 0.7 to 84 m/day for a low-centered polygon and 0.01 to 0.3 m/day for a high-centered polygon. Wales et al. (2020) stated that these are considered lower bound estimates of horizontal conductivity since, based solely on breakthrough times, the approach is unable to isolate horizontal and vertical flow effects on arrival times. Beckwith et al. (2003) perform laboratory measurements of anisotropy on small samples from peat soils where horizontal conductivities were around 6-7 m/day and vertical conductivities were around 0.2-0.4 m/day indicating preferential horizontal flow. The soil samples used in Beckwith et al. (2003) were from peat bogs in England, and therefore would not have been subjected to freeze-thaw dynamics that current ice-wedge polygons will have been subjected



to in recent times, which would presumably lead to even greater preferential horizontal flow. Therefore, these estimates may provide a lower bound estimate for Arctic peat horizontal hydraulic conductivity. Based on the existing literature and above

considerations, we considered hydraulic conductivities from 0.005-5 m/day, primarily focusing on anisotropic values from 0.1 to 100. These values allow us to investigate the effects of anisotropy on the drainage pathways and timing using hydraulic conductivities consistent with currently available measurements.

## 2.2  Model overview

We use recently developed 3D-axisymmetric analytical solutions derived in Zlotnik et al. (2020), and briefly described in

Appendix A, of hydraulic heads and the stream function in the thawed soil layer below a polygon center (dashed rectangle in Figure 1) to investigate inundated low-centered polygon drainage pathways. In order to generalize the results with regard to polygon center and trough ponded heights, we use non-dimensional forms of the hydraulic head and stream function (refer to equation A6 and A7). The non-dimensional hydraulic heads and stream function are independent of time, representing the relative pattern of hydraulic heads and stream function throughout the drainage process. The dimensional values at different

times would indicate the change in absolute magnitude of these patterns as the drainage process proceeds. In order to evaluate drainage pathways, we plot flow nets, as illustrated in Figure 1, composed of lines of equal non-dimensional hydraulic head ($h^*(r^*, z^*)$), referred to as equipotentials, and contours of the stream function ($\Psi^*(r^*, z^*)$) as a function of radius and depth. The stream function contours are referred to as *streamlines* and are used here to identify steady drainage pathways.

Non-dimensional hydraulic head lines are drawn from 0.05 to 0.95 by increments of 0.1. Dimensional heads ($h(r, z, t)$) can

be obtained from the non-dimensional heads ($h^*(r^*, z^*)$) using the ponded height of water in the polygon center $H_c(t)$ and the height of water in the trough $H_t$ as

$$h(r, z, t) = H_t + (H_c(t) - H_t) h^*(r^*, z^*), \; r^* = \frac{r}{L} \sqrt{\frac{K_z}{K_r}}, \; z^* = \frac{z}{L}, \tag{1}$$

where $r^*$ and $z^*$ are non-dimensional radius and depth, respectively, $L$ is the polygon thaw depth, and $K_r$ and $K_z$ are the radial (horizontal) and vertical hydraulic conductivity, respectively (refer to Figure 1).

We normalize the stream function $\Psi(r^*, z^*)$ from 0 to 1, and similar to heads, plot stream function contours (streamlines) from 0.05 to 0.95 by increments of 0.1. Given this set of streamlines, the region between any two adjacent streamlines conveys 10% of the drainage, while the two remaining regions (less than 0.05 and greater than 0.95), convey 5% of the drainage each. In order to quantitatively evaluate the spread of drainage within the polygon, we calculate the percent of each polygon volume accessed by 95% of the drainage (polygon volume with non-dimensional stream function > 0.05).

The change in non-dimensional ponded water height in the polygon center due solely to drainage (the depletion curve) over time can be expressed by a simple exponential decay function as

$$H_c^*(t) = e^{-t/t_L} \tag{2}$$





where $t_L$ is the characteristic time of drainage (refer to equation A10), defined as the time when the ponded height is $1/e$ times, or $\sim 37\%$ of, its original height $H_{c,0}$. The dimensional ponded height $H_c(t)$ can be obtained as

$$H_c(t) = H_t + (H_{c,0} - H_t)H_c^*(t). \tag{3}$$

## 3 Results

Here, we present drainage flow nets for various aspect ratios and anisotropies, polygon center ponded height depletion curves, and maps of the percent of the thawed soil accessed by 95% of the drainage flow and depletion characteristic times as a function of aspect ratio and anisotropy. We compare the effect of aspect ratio on drainage pathways in an isotropic and a highly anisotropic ($K_r/K_z = 100$) polygon. The drainage pathways are also evaluated for various anisotropies holding the aspect ratio constant. A global perspective of the combined effects of aspect ratio and anisotropy on the focusing/spreading of the drainage flow is provided by mapping the percent of the polygon thawed soil volume that is accessed by 95% of the drainage flow. We evaluate the effect of aspect ratio and anisotropy on drainage time by plotting the polygon ponded water height depletion curves. A global perspective on the combined effects of aspect ratio and anisotropy on drainage time is presented by mapping the depletion characteristic time. We illustrate the counteracting effects of aspect ratio and anisotropy on drainage pathways by showing two mathematically equivalent solutions of drainage.

### 3.1 Drainage pathways

The effect of aspect ratio on drainage pathways when the hydraulic conductivities are isotropic ($K_r/K_z = 1$) is shown in Figure 2. Under isotropic hydraulic conductivities, results show that the drainage pathway for high aspect ratio polygons will be predominantly isolated to an annular region at the periphery of the polygon center. As the aspect ratio decreases (in other words, the thawed region of the polygon subsurface becomes deeper with respect to its width), the portion of the polygon accessed by drainage increases. For an aspect ratio of 20, 95% of the drainage is focused within around 5% of the polygon volume, while for an aspect ratio of 2.5, the drainage is spread over around 29% of the polygon volume. The spreading (increase in the accessed volume) occurs along the radial direction, where the horizontal extent of the accessed volume moves towards the middle of the polygon center ($r = 0$), while the vertical extent is nearly unchanged. The results in Figure 2 indicate that throughout the thaw season, as the thaw depth increases, the drainage path will spread out towards the middle of the polygon center. Similarly, Figure 2 can be used to evaluate drainage pathways of polygons of different widths but similar thaw depths. In this context, Figure 2 indicates that wider polygons will have more focused drainage, while drainage for the small polygons will be more dispersed.

The effect of anisotropy on drainage pathways when the aspect ratio is held constant ($R/L = 10$) is shown in Figure 3. As anisotropy increases (in other words, the more that horizontal conductivity dominates), the region accessed by drainage flow becomes larger. Similar to a decreasing aspect ratio in Figure 2, increasing anisotropy leads to a larger radial extent of the accessed region, while the vertical extent is nearly unaffected. When the vertical conductivity is ten times the horizontal (top plot), only around 3% of the polygon volume is accessed by 95% of the drainage. If horizontal conductivity is 100 times

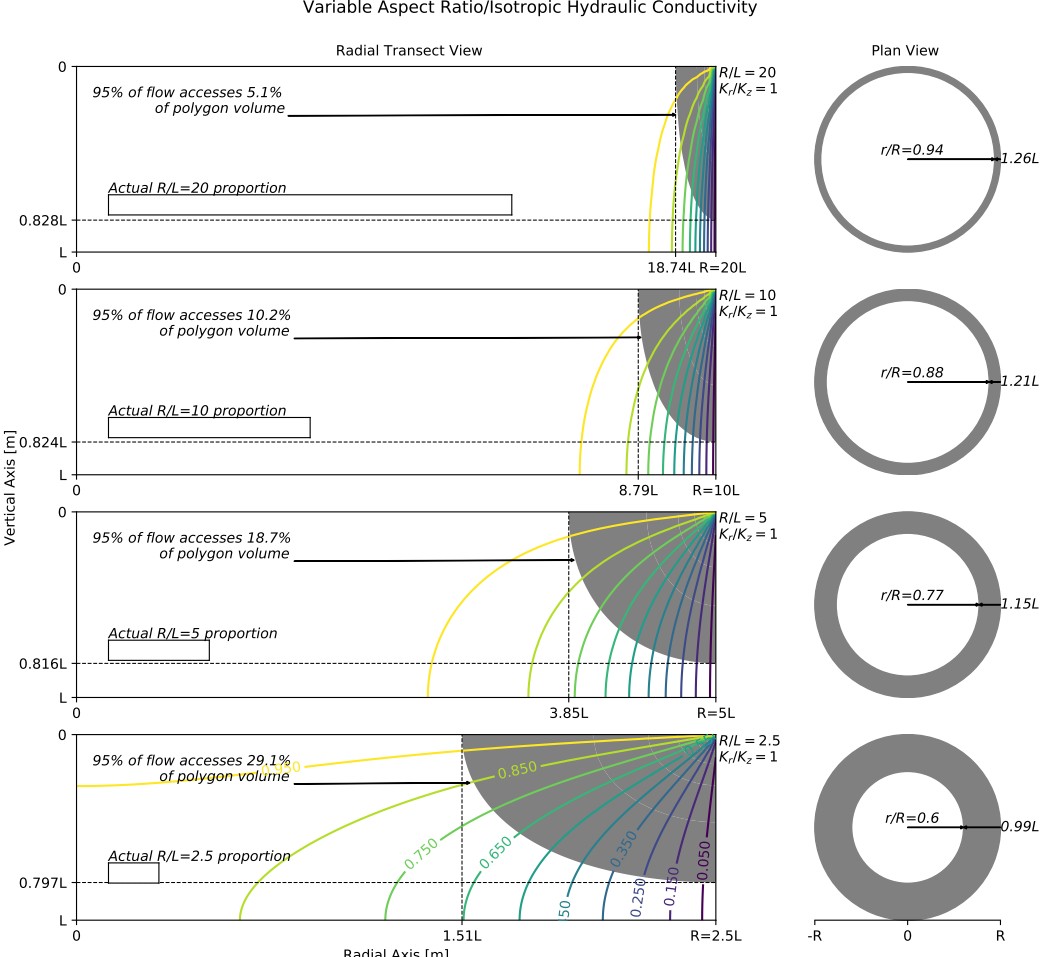

**Figure 2.** Effect of polygon aspect ratio on polygon drainage with isotropic hydraulic conductivity. Plots along the left contain polygon radial transect head contours (colored lines) and filled stream tubes (grey regions) for several polygon aspect ratios (radius/thickness). The grey shaded region denotes the portion of the transect accessed by 95% of the flow. The plots along the right contain corresponding grey rings indicating the surface area where 95% of the polygon flow infiltrates. Each plot along the left contains a rectangle drawn to the actual proportions for the given polygon aspect ratio. In all cases, the anisotropy (horizontal/vertical conductivity) is fixed at unity.

vertical conductivity (bottom plot), around 49% is accessed. These results indicate that anisotropy has a significant impact on ice-wedge polygon drainage pathways.
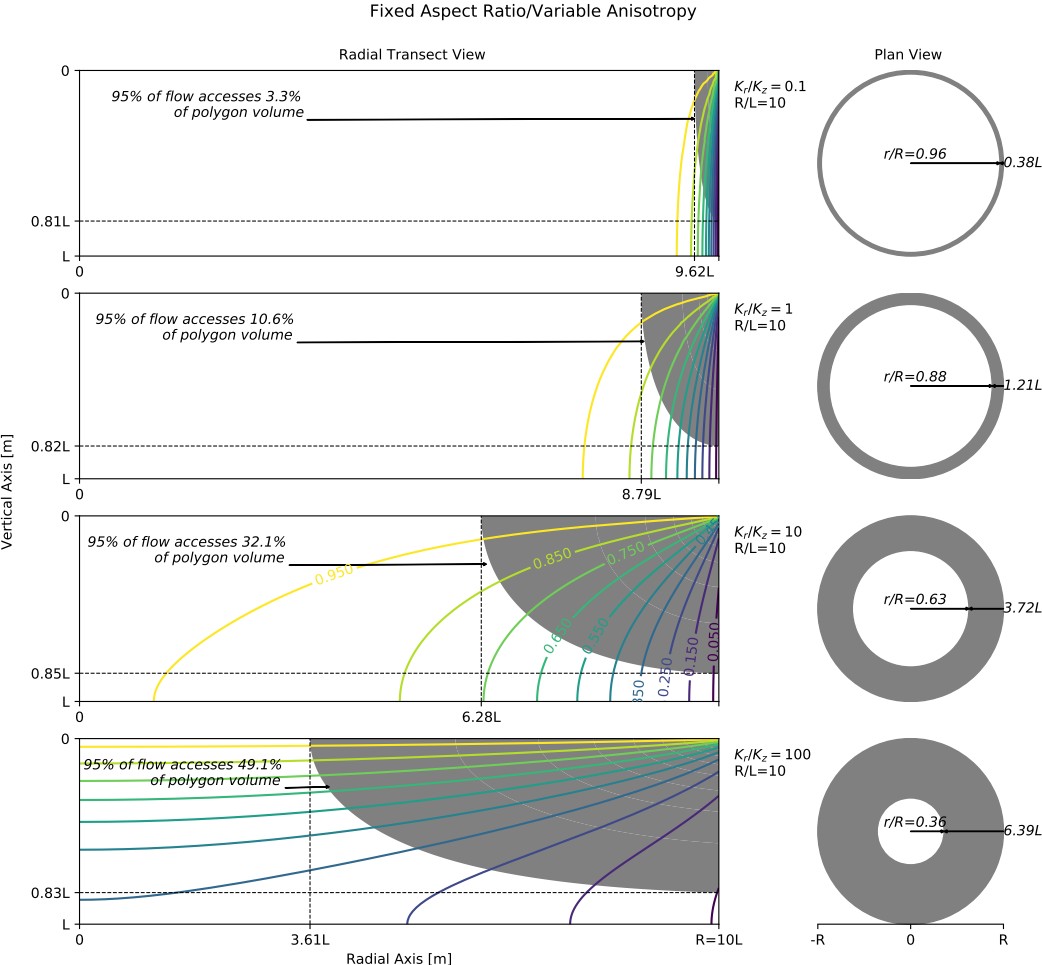

**Figure 3.** Effect of hydraulic conductivity anisotropy on polygon drainage. Plots along the left contain polygon radial transect head contours (colored lines) and filled stream tubes (grey regions) for several anisotropies (horizontal/vertical conductivity). The grey shaded region denotes the portion of the transect accessed by 95% of the flow. The plots along the right contain corresponding grey rings indicating the surface area where 95% of the polygon flow infiltrates. Each plot along the left contains a rectangle drawn to the actual proportions for the given polygon aspect ratio. In all cases, the polygon aspect ratio (radius/thickness) is fixed at 10.

The effect of aspect ratio on drainage pathways when the hydraulic conductivities are highly anisotropic ($K_r/K_z = 100$) is shown in Figure 4. In this case, contrary to the isotropic case in Figure 2, as the aspect ratio decreases, the accessed volume generally decreases. However, the largest accessed volume is for an aspect ratio of 10, not 20. This nuance in the dependence of

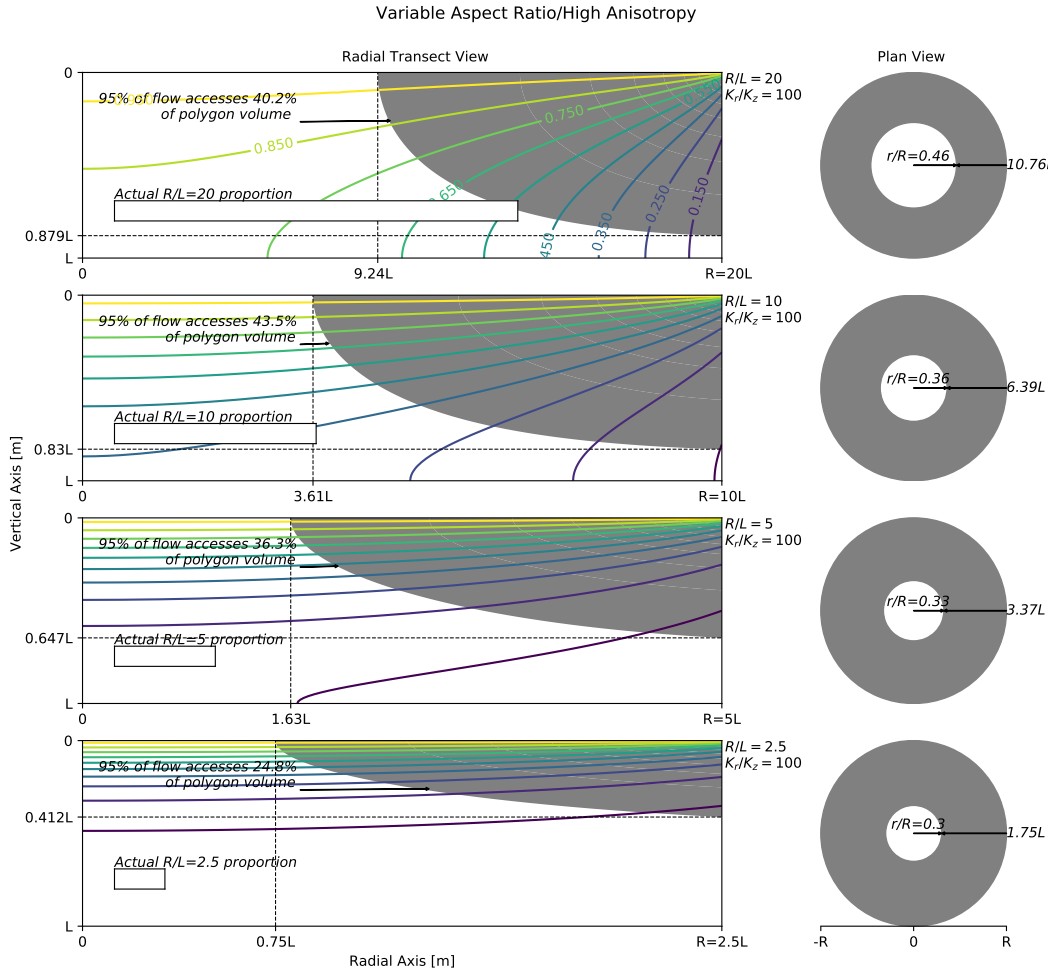

**Figure 4.** Effect of polygon aspect ratio on polygon drainage with high hydraulic conductivity anisotropy ($K_r/K_z = 100$). Plots along the left contain polygon radial transect head contours (colored lines) and filled stream tubes (grey regions) for several polygon aspect ratios (radius/thickness). The grey shaded region denotes the portion of the transect accessed by 95% of the flow. The plots along the right contain corresponding grey rings indicating the surface area where 95% of the polygon flow infiltrates. Each plot along the left contains a rectangle drawn to the actual proportions for the given polygon aspect ratio. In all cases, the anisotropy (horizontal conductivity/vertical conductivity) is fixed at unity.

accessed volume to aspect ratio with high anisotropy is due to the competing effects of radial extension and vertical contraction



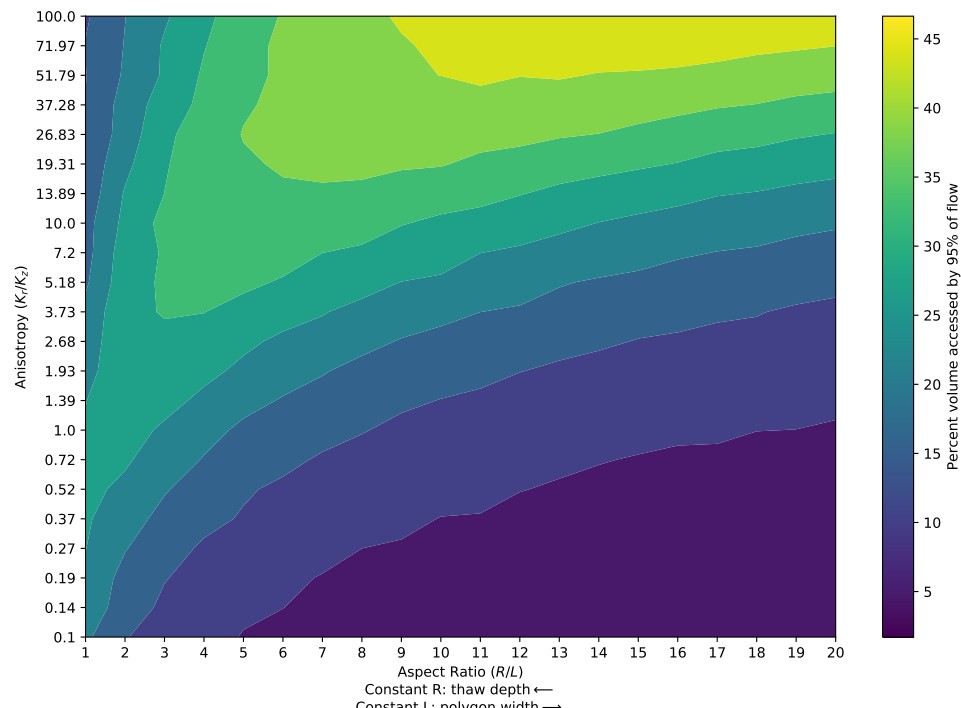

**Figure 5.** Contour map of the percent of the polygon access by 95% of the flow as a function of polygon aspect ratio and anisotropy. By choosing a constant value for $R$ $(L = R/x)$, horizontal transects parallel to the x-axis represent the effect of thaw depth increasing from right to left, while they represent increasing polygon width from left to right for constant values of $L$ $(R = Lx)$.

of the accessed volume as aspect ratio decreases. By comparing Figures 2 and 4, it is also apparent that the volume accessed by 95% of the drainage is generally larger with higher anisotropy.

To gain a global perspective on the trends in drainage pathways with aspect ratio and anisotropy, Figure 5 maps the percent of the polygon accessed by 95% of the drainage as a function of aspect ratio and anisotropy. This approach illustrates the

overall structure of the combined effect of aspect ratio and anisotropy on accessed volume (focused versus dispersed drainage). The trend in accessed volume with respect to aspect ratio in Figure 2 is represented along the anisotropy=1 transect in Figure 5, while the trend in Figure 4 is represented by the anisotropy=100 transect. There is a spreading ridge-like structure in the accessed volume with increasing aspect ratio and anisotropy with decreasing accessed volume on either side of this curved ridge line. The ridge represents the optimal balance of radial and vertical extension of the accessed volume region, as described

in the previous paragraph.

This structure explains the increasing accessed volume with decreasing aspect ratio in Figure 2 and the maximum accessed volume at aspect ratio of 10 in Figure 4. Similarly, the trend in accessed volume with respect to anisotropy is represented by

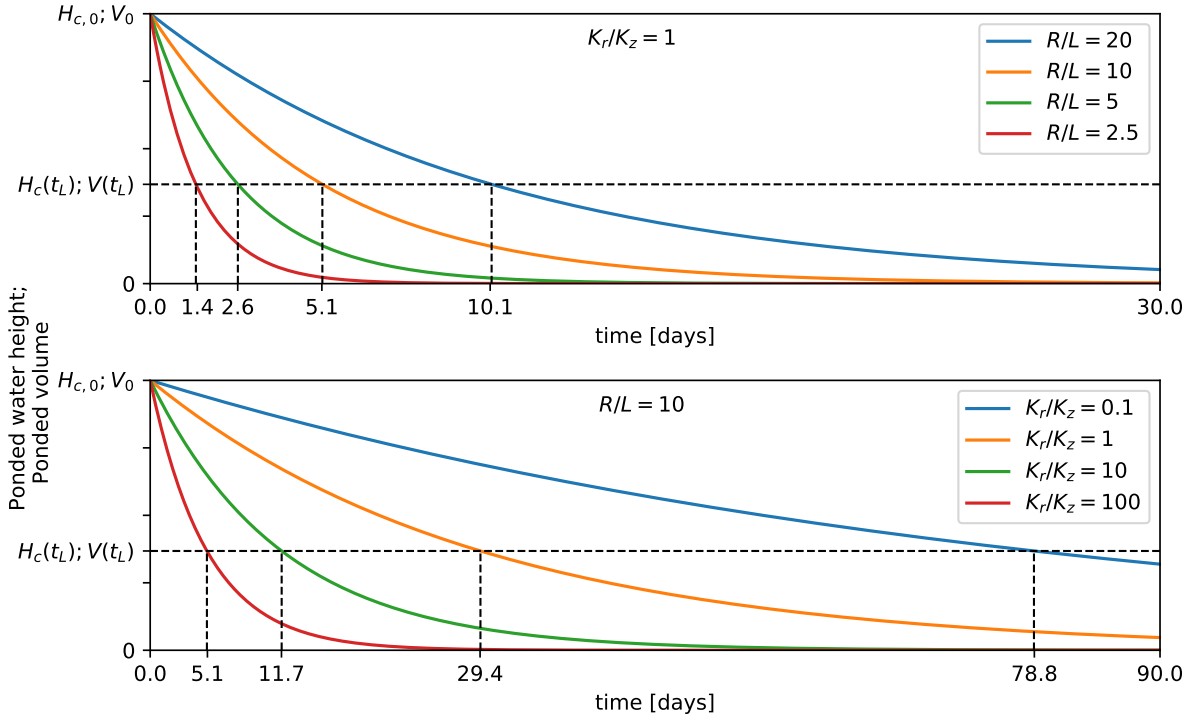

**Figure 6.** Depletion of ponded water height over time ($H_c(t)$) for alternative polygon aspect ratios (radius/depth ($R/L$); top plot) and anisotropy ($K_r/K_z$; bottom plot). In the top plot, the anisotropy is fixed at unity and in the bottom plot the aspect ratio is fixed at 10. The curves also describe the depletion of ponded water volume over time ($V(t)$). Dashed lines indicate the characteristic times ($t_L$) in each case.

the aspect ratio=10 transect in Figure 3. The drainage flow is most spread out (largest accessed volume) when the aspect ratio is large and the anisotropy is high (yellow region in the upper left of Figure 5). The drainage is the most focused (least accessed
volume) when the aspect ratio is large and the anisotropy is low (dark blue region in the lower left of Figure 5).

### 3.2   Ponded water depletion

Depletion curves for the non-dimensional ponded water height in the polygon center for various aspect ratios and anisotropies are shown in Figure 6. Note that since the depletion of the volume of ponded water is directly related to the ponded water height, the plots in Figure 6 can be used to obtain either, as indicated by the y-axis labels. As a point of reference between
depletion curves, the characteristic time (the time when the height or volume of ponded water reaches $1/e \approx 0.37$ its initial height or volume) is indicated.

It is apparent from these plots that small, deeply thawed (lower aspect ratio) polygons will drain faster than wide, shallowly thawed (high aspect ratio) polygons, while polygons with higher anisotropy will drain faster than those with low anisotropy. It is also apparent that the increase in drainage time with aspect ratio is nearly linear, while for anisotropy, it is exponential.

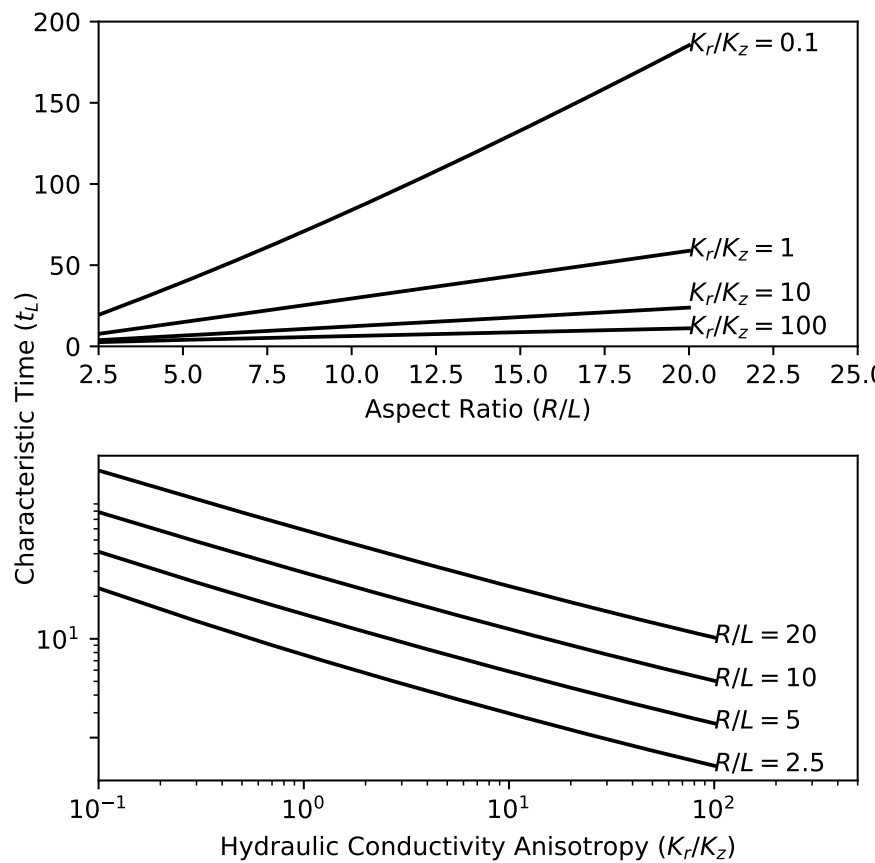

**Figure 7.** Characteristic time as a function of polygon aspect ratio (top plot) and anisotropy (bottom plot). Note that the bottom plot has log transformed axes.

This is further illustrated in Figure 7, where we plot the trend in characteristic times as a function of aspect ratio for various anisotropies (top plot) and as a function of anisotropies for various aspect ratios (bottom plot). It is apparent that drainage time (represented by characteristic time) increases in a nearly linear fashion, particularly for high anisotropies, with slight upward curvature increasing for low anisotropies. The drainage time decreases in a nearly log-log fashion (exponentially) with increasing anisotropy with an identical trend for different aspect ratios.

A global perspective on drainage timing trends where characteristic time is mapped as a function of aspect ratio and anisotropy is shown in Figure 8. The fastest (shortest) drainage times are achieved with low aspect ratio and high anisotropy, while the slowest (longest) drainage times are achieved with high aspect ratio and low anisotropy. These trends indicate that given the same thawed soil layer thickness, wider polygons will drain more slowly than small polygons. Or, for polygons with similar aspect ratio, those with preferential horizontal flow will drain most quickly, while those with preferential vertical flow

will drain most slowly.

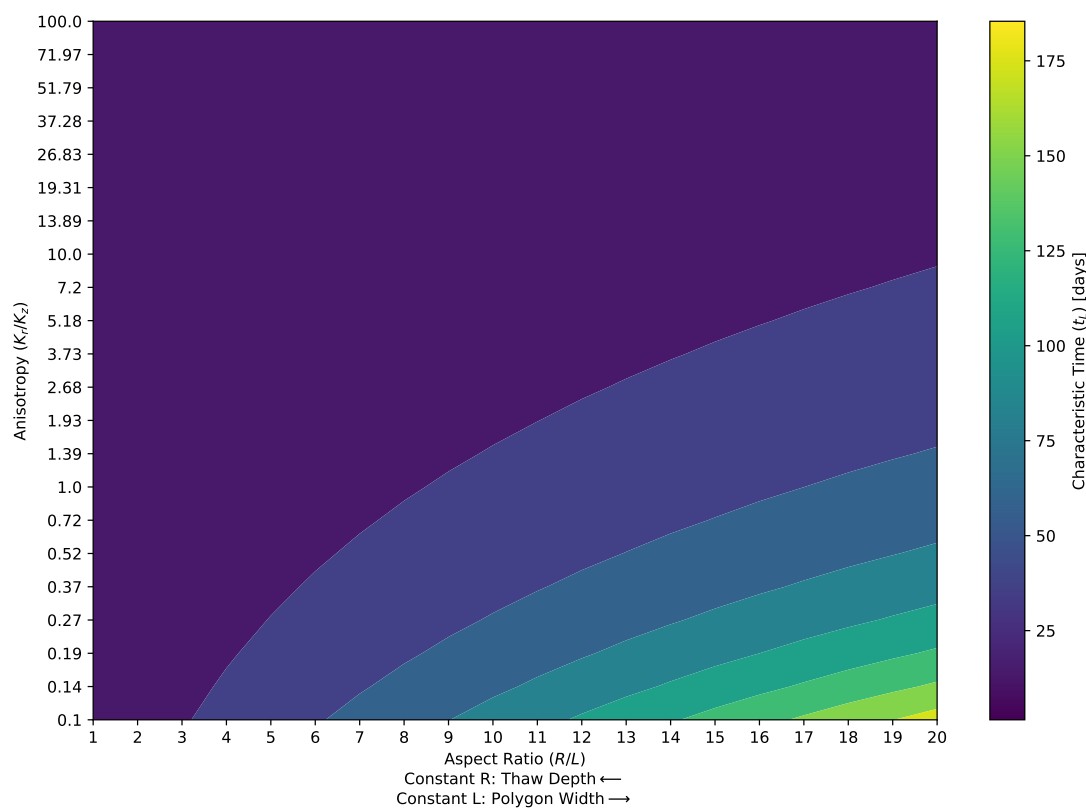

**Figure 8.** Contour map of the characteristic time of polygon drainage as a function of polygon aspect ratio and anisotropy. By choosing a constant value for $R$ ($L = R/x$), horizontal transects parallel to the x-axis represent the effect of thaw depth increasing from right to left, while they represent increasing polygon width from left to right for constant values of $L$ ($R = Lx$).

### 3.3 Counteracting effect of aspect ratio and anisotropy

It is important to note that aspect ratio and anisotropy have similar effects on drainage. Within the model used here, in a similar fashion to aspect ratio, anisotropy stretches the domain by multiplying the non-dimensional radius by $\sqrt{(K_z/K_r)}$ (refer to definition of $r^*$ in equation 1). As a result, the effect of increasing anisotropy in the analytical solution is mathematically

equivalent to a decrease in aspect ratio. In the end, the equipotential heads and streamlines computed on stretched or compressed radial coordinates ($r^*$) are assigned back to non-modified radial coordinates ($r$). This would lead one to believe that it should be possible to obtain two scenarios with different aspect ratios and anisotropies that produce the same mathematical solution. For example, doubling the aspect ratio should produce the identical effect as dividing the anisotropy by four. However, this is not the case because the solution involves a Biot parameter (Bi) which defines a ratio of the ability for fluid to conduct across
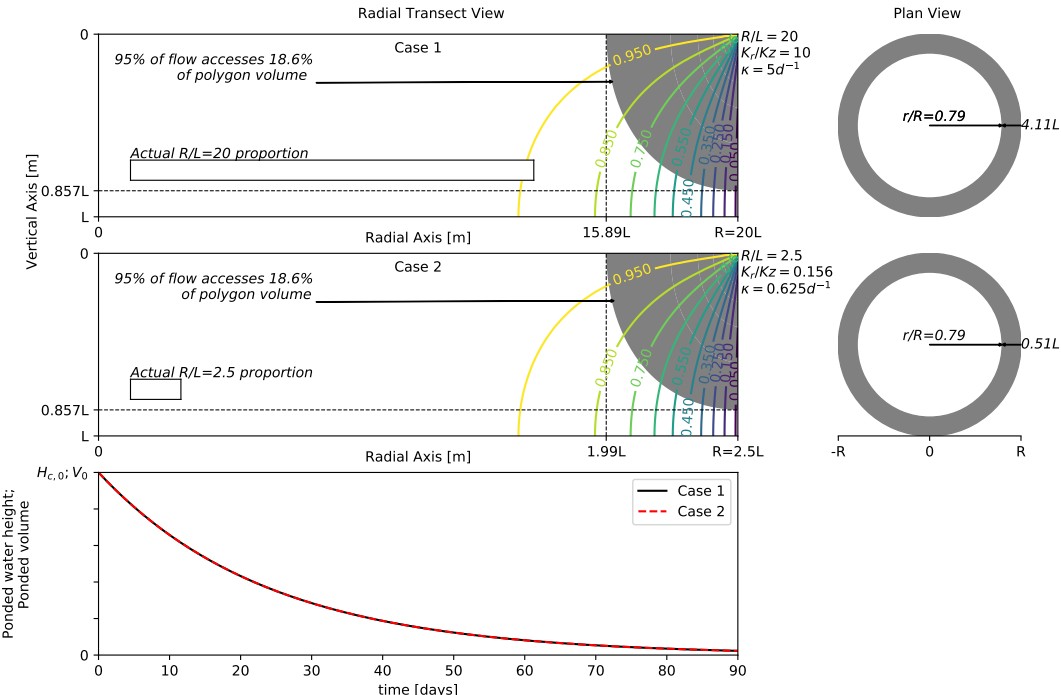

**Figure 9.** Attaining the same mathematical solution by modifying polygon aspect ratio ($R/L$), anisotropy ($K_r/K_z$), and rim conductance ($\kappa$). The two upper plots along the left contain polygon radial transect head contours (colored lines) and filled stream tubes (grey regions) for several polygon aspect ratios (radius/thickness). The grey shaded region denotes the portion of the transect accessed by 95% of the flow. The two upper plots along the right contain corresponding grey rings indicating the surface area where 95% of the polygon flow infiltrates. Each plot along the left contains a rectangle drawn to the actual proportions for the given polygon aspect ratio. The ponded water height/volume depletion curves for both cases are presented in the bottom plot.

the drainage interface (the vertical outer boundary) relative to the internal hydraulic conductivity as

$$\mathrm{Bi} = \frac{\kappa L}{\sqrt{K_r K_z}}, \tag{4}$$

where $\kappa$ characterizes the resistance to drainage across the outer vertical boundary of the model (refer to Appendix A for more details). A large Bi indicates that the outlet boundary interface of the model will not limit the drainage, while a small Bi will result in outlet boundary interface limited drainage. Therefore, to obtain an identical mathematical drainage solution using

different combinations of aspect ratio and anisotropy, one would also need to ensure that Bi is not modified in the process. However, given two solutions, this requires satisfying the conflicting conditions that $K_{z1}/K_{r1} = K_{z2}/K_{r2}$ and $K_{r1}K_{z1} = K_{r2}K_{z2}$ (refer to equations 1 and 4), where subscripts 1 and 2 refer to the two solutions. Since these two constraints cannot be simultaneously met, $\kappa$ or $L$ must also be modified along with aspect ratio and anisotropy to achieve an identical mathematical



solution of drainage. An example where identical mathematical solutions for drainage are obtained is provided in Figure 9,

where the bottom plot is obtained by taking the properties of the top plot and dividing the aspect ratio by 8, dividing the anisotropy by $8^2$, and dividing $\kappa$ by 8. While the solutions are mathematically identical, note that the axes in both plots are not to scale (actual proportions are provided as insets to the plots) and that there are differences in some relative dimensions indicated in the figure (for example, the relative width of the polygon volume accessed by 95% of the drainage flow is around $4L$ in the top case and around $0.5L$ in the bottom case).

In the bottom plot in Figure 9, we demonstrate that despite these differences, the depletion curves are identical for both cases. This result is due to our choice to modify $R$ to change the aspect ratio and modify $K_r$ to change the anisotropy. If either $L$ or $K_z$ are chosen to modify the aspect ratio or anisotropy, respectively, the depletion curves would not necessarily by identical. For example, if $L$ is multiplied by 8, $K_z$ multiplied by $8^2$, and $\kappa$ divided by 8 (equivalent modifications to aspect ratio and anisotropy as above), the drainage flow net would still be mathematical equivalent, but the characteristic time would be 8 times

shorter than in the original case (refer to equation A10). Therefore, while it is possible to obtain mathematically equivalent drainage patterns with counteracting modifications to aspect ratio, anisotropy, and outflow resistance, the temporal depletion will only be equivalent if $R$ and $K_r$ are used to modify the aspect ratio and anisotropy, respectively.

## 4    Discussion

Our analysis provides new insights into the manner in which geometry and anisotropy effect how inundated ice-wedge polygons

retain and slowly release water from their centers to their troughs, which form the drainage network of polygonal tundra landscapes. Using a mathematical representation of inundated ice-wedge polygon drainage (Zlotnik et al., 2020) based on extensive field observations (Helbig et al., 2013; Koch et al., 2014; Liljedahl and Wilson, 2016; Wales et al., 2020), we quantify the sensitivity of inundated ice-wedge polygon drainage to representative polygon sizes, inter- and intra-annual changes in thaw depth, and preferential flow (hydraulic conductivity anisotropy).

A key result of this study is that the geometry and anisotropy of the polygon subsurface have a significant effect on the region of the polygon subsurface predominantly accessed by drainage flow and the time to transition from inundated to drained. Due to the geometry of inundated polygons, the primary drainage pathway is restricted to an annular, radially-peripheral region of the ice-wedge polygon center near the rim. As a result, the middle and lower portions of the polygon center are excluded from the majority of the drainage to varying degrees depending on the aspect ratio and anisotropy. In addition, the majority

of the ponded water will flow to an outer ring of the polygon before infiltrating into the subsurface. Field observations have indicated not only the existence of intra-polygon biogeochemical diversity (Zona et al., 2011; Newman et al., 2015), but that, based on mineral and nutrient loading, pervasive subsurface flow from centers to troughs exists (Koch et al., 2014). Based on our analysis, polygon geometry and anisotropy will have important effects on the biogeochemistry of polygons (Heikoop et al., 2015; Throckmorton et al., 2015; Newman et al., 2015; Wales et al., 2020; Plaza et al., 2019) effecting dissolved organic matter

and mineral flushing from the subsurface of polygon centers to the surface waters of troughs. This is important not only simply





with regard to discharge of dissolved organic matter from polygonal tundra landscapes, but also for the biogeochemical effects due to photolysis of dissolved organic matter newly exposed to sunlight (Laurion and Mladenov, 2013; Cory et al., 2014).

The potential that the annular, radially-peripheral region near the rims will be well flushed of nutrients while the middle may not indicates the need for additional field studies designed to measure the effects of anisotropy and preferential flow paths on thermal-hydrology and biogeochemistry. For isotropic cases, it should also be considered that the drainage will spread out further towards the middle of the polygon center as the thaw season progresses and the thawed soil layer thickens (in other words, the aspect ratio decreases; Figure 2 and 4). However, with high anisotropy, the drainage still spreads out towards the polygon middle as the thaw season progresses (aspect ratio decreases), but the depth of the main drainage pathways contract towards the ground surface (Figure 4). Ultimately, the effect of this vertical contraction outweighs the lateral spreading, leading to a smaller region being accessed by drainage. Therefore, in the case of low aspect ratio and high anisotropy, the dissolved organic matter and nutrients closer to the surface will be flushed more than the deeper stores of dissolved organic matter and nutrients. Deeper nutrients that become available for transport because of recent thaw may still be less accessible to aqueous transport due to drainage pathways in polygons with high anisotropy. This research reinforces the need for field studies on anisotropy and preferential flow in polygon landscapes in order to better understand the hydrologic transitions and feedbacks that will occur in a warming climate.

The drainage pathways and timing presented here are based on fundamental hydrogeological principles (Harr, 1962; Cedergren, 1968; Freeze and Cherry, 1979; Bear, 1979) using a generalization of ice-wedge polygon geometry and hydraulic properties that allow fundamental insights of generalized drainage pathways and timing of inundated ice-wedge polygons to be obtained. Our solutions capture the fundamental physical forces that ponded water exerts on a cylindrical porous disc with drainage allowed radially through its sides. While the cylindrical geometry and anisotropy considered here do not cover all potential variations present in ice-wedge polygons, the impact of those variations will cause deviations around the base case, idealized scenarios considered here. For example, anomalous heterogeneities will warp the base case hydraulic head equipotentials and stream lines. The cylindrical idealization of ice-wedge polygon geometry will best approximate drainage in nearly symmetrical hexagonal ice-wedge polygons, while non-symmetric, square ice-wedge polygon drainage will deviate most from that shown here.

As the analytical solution applies to ponded conditions, it is not applicable to freeze-up at the end of the thaw season after the ponded water in the ice-wedge polygon center freezes. At this time, the active layer freezes simultaneously from the top and bottom and cryosuction draws water towards the freezing front. This redistribution of water affects ice-wedge polygon drainage and Wales et al. (2020) postulate that it may explain some of the observations of their tracer test. More complex models than applied here are required to capture the details of water redistribution during freeze-up (Painter, 2011; Atchley et al., 2016; Schuh et al., 2017).

Ice-wedge polygons with focused, deep annular flow towards their periphery may also direct warmer surface waters towards the top of ice-wedges, resulting in enhanced ice-wedge degradation (Wright et al., 2009). Based on our calculated drainage pathways, advective heat transport will be most pronounced in large aspect ratio polygons with low anisotropy (refer to the top plots in Figures 2 and 3). Therefore, drainage events for wide, isotropic (or with preferential vertical flow) polygons may result





in enhanced ice-wedge top thawing, which would promote low- to high-centered polygon transition. The more spread out the drainage is throughout the thawed soil layer, the less pronounced this effect may be (for example, wide polygons with high anisotropy as in the upper plots in Figure 4). However, small polygons with high anisotropy will restrict the drainage flow near the ground surface, potentially reducing the advective transport of heat to the ice-wedge top as well (bottom plot in Figure 4).

Small polygons with deeply thawed soil layers (low aspect ratios) and high horizontal preferential flow (high anisotropy) have the potential to drain most quickly. Therefore, all other factors being equal, regions of polygonal tundra characterized by small, deeply thawed, anisotropic polygons will drain more quickly, and consequently would have a greater potential for nutrient flushing, transition from methane to carbon dioxide atmospheric emissions, and biological succession than regions with large, shallowly thawed, isotropic polygons. Of course, for a given location, other factors such as regional flow patterns,

large scale topography, etc. will influence the regions overall drainage timing. However, along with these other factors, our analysis indicates that aspect ratio will have a nearly linear positive relationship while anisotropy will have an exponential negative relationship with drainage timing.

## 5   Conclusions

- The majority of drainage from inundated ice-wedge polygon centers occurs along an annular region along their radial
periphery; however, polygon aspect ratio and hydraulic conductivity anisotropy significantly impact the drainage pathways.

  - A combination of high aspect ratios (wide, shallow polygons) and high anisotropy (preferential horizontal flow) result in the greatest spreading of drainage flow and largest fraction of the polygon volume being accessed by drainage flow.

- A combination of high aspect ratios (wide, shallow polygons) and low anisotropy (preferential vertical flow) result in the greatest focusing of drainage flow and smallest fraction of the polygon volume being accessed by drainage flow.

  - Combinations of aspect ratio and anisotropy have counteracting effects of radial versus vertical extension/contraction of drainage pathways, producing non-monotonic relationships between aspect ratio/anisotropy and accessed vol-
ume (ridgeline in accessed volume response surface in Figure 5).

- The characteristic time for a polygon to drain has an approximately positive linear relationship with aspect ratio; in other words, wide, shallow polygons drain slowly while small, deep polygons drain quickly.

- The characteristic time for a polygon to drain has a negative exponential relationship with anisotropy; in other words, preferential horizontal flow leads to exponentially faster drainage.

*Code availability.* Matlab scripts of the analytical solutions are provided as supplementary information.



## Appendix A

Here, we present the analytical solutions for hydraulic heads and stream function under the center of an inundated ice-wedge polygon and the depletion curve of the ponded water height due to drainage to the surrounding trough. For details on the derivations, refer to Zlotnik et al. (2020).

### A1 Hydraulic head and stream function analytical solutions

We idealize the subsurface below the center region of a low-centered polygon as a thin cylinder with radius in the horizontal direction and length in the vertical direction (refer to Figure 1). Based on this approximation, the initial (steady-state) hydraulic heads ($h(r,z,0)$) in an inundated polygon will satisfy the following equation:

$$\frac{K_r}{r}\frac{\partial}{\partial r}\left(r\frac{\partial h}{\partial r}\right) + K_z\frac{\partial^2 h}{\partial z^2} = 0, \tag{A1}$$

where $h$ is the hydraulic head, $r$ is the radial coordinate, $z$ is the depth coordinate (positive in the downward vertical direction), $K_r$ is the horizontal (radial) hydraulic conductivity, and $K_z$ is the vertical hydraulic conductivity. The system is fully specified by ensuring that (1) the heads along the central vertical axis ($h(0,z,t)$) are finite, (2) the heads along the ground surface are equal to the height of ponded water in the polygon center ($h(r,0,t) = H_c(t)$), (3) the change in heads along the outer vertical boundary are governed by the change in heads along the boundary and the height of water in the trough ($H_t$)

$$-K_r\frac{\partial h(R,z,t)}{\partial r} = \kappa(h(R,z,t) - H_t), \tag{A2}$$

(4) the bottom of the model has a zero head gradient

$$\frac{\partial h(r,L,t)}{\partial z} = 0,\ 0 < r < R, \tag{A3}$$

and (5) the change in volume of ponded water in the polygon center is related to the vertical head gradients along the ground surface,

$$\pi R^2\frac{dH_c(t)}{dt} = 2\pi K_z\int_0^R \frac{\partial h(r,0,t)}{\partial z}r\,dr,\ H_c(0) = H_{c,0}, \tag{A4}$$

where $R$ is the radius of the polygon, $L$ is the depth of the thawed subsurface of the polygon, $H_c(t)$ is the ponded water height in the center of the polygon at time $t$, $H_t$ is the height of water in the trough, and $\kappa$ characterizes the resistance to flow across the drainage interface to the trough (in our case, the hydraulic resistance of the soil layer under the rim).

Using dimensionless coordinates and parameters defined as

$$r^* = \frac{r}{L}\sqrt{\frac{K_z}{K_r}},\ \ z^* = \frac{z}{L},\ \ R^* = \frac{R}{L}\sqrt{\frac{K_z}{K_r}},\ \ \text{Bi} = \frac{\kappa L}{\sqrt{K_r K_z}}, \tag{A5}$$

dimensionless solutions for hydraulic heads and the stream function can be obtained as

$$h^*(r^*,z^*) = \frac{h(r,z,t) - H_t}{H_c(t) - H_t} = 2\sum_{n=1}^{\infty}\frac{J_1(\lambda_n R^*)}{\lambda_n R^*\left[J_0^2(\lambda_n R^*) + J_1^2(\lambda_n R^*)\right]}\frac{\cosh(\lambda_n(1 - z^*))}{\cosh(\lambda_n)}J_0(\lambda_n r^*) \tag{A6}$$



and

$$\Psi^*(r^*, z^*) = \frac{\psi(r^*, z^*)}{\psi(R^*, 0)} = 2 \sum_{n=1}^{\infty} \frac{J_1(\lambda_n R^*) J_1(\lambda_n r^*) r^*}{\lambda_n R^* [J_0^2(\lambda_n R^*) + J_1^2(\lambda_n R^*)]} \frac{\sinh(\lambda_n(1 - z^*))}{\cosh(\lambda_n)}, \tag{A7}$$

respectively, where $J_m$, with $m=0$ or 1, is the Bessel function of the first kind of $m$th order and $\lambda_n$ is the $n$th root of the equation

$$\lambda_n J_1(\lambda_n R^*) = \mathrm{Bi} J_0(\lambda_n R^*), \ n = 1, 2, \ldots \tag{A8}$$

The solutions can be verified by direct substitution of equations A6 and A7 into the boundary value problem defined by equations A1 to A5.

## A2   Ponded height depletion curve analytical solution

The ponded height depletion curve can be defined as

$$H_c(t) = H_t + [H_{c,0} - H_t]e^{-t/t_L}, \tag{A9}$$

where $t_L$ is the characteristic depletion time defined as

$$t_L = \frac{R^2}{2 K_r L F(\mathrm{Bi}, R^*)}, \tag{A10}$$

which is the time when $H_c(t)/H_{c,0} = 1/e$; in other words, when the ponded height is approximately 37% of its initial. The function $F(\mathrm{Bi}, R^*)$ can be evaluated as

$$F(\mathrm{Bi}, R^*) = \int_0^{R^*} \frac{\partial h^*(r^*, 0)}{\partial z^*} r^* dr^* = 2 \sum_{n=1}^{\infty} \frac{\tanh(\lambda_n)}{\lambda_n \left[(\lambda_n/\mathrm{Bi})^2 + 1\right]}. \tag{A11}$$

The depletion curve can be expressed in non-dimensional terms as

$$H^*(t^*) = \frac{H_c(t) - H_t}{H_{c,0} - H_t} e^{-t^*}, \tag{A12}$$

where non-dimensional time $t^* = t/t_L$. The solution can be verified by direct substitution of equation A9 into equation A4.

*Author contributions.*   DH and VZ developed the conception model of inundated polygonal tundra hydrology. VZ derived the analytical solutions. VZ and DH encoded the analytical solutions. CA created Figure 1. DH created Figures 2-9. AA provided text for the Introduction and Discussion sections. AA, BN, and CW provided critical reviews of the manuscript. CW secured funding. TBD...

*Competing interests.*   The authors declare no competing interests.



*Acknowledgements.* The Next Generation Ecosystem Experiments Arctic (NGEE-Arctic) project (DOE ERKP757), funded by the Office of Biological and Environmental Research within the U.S. Department of Energy's Office of Science supported this research. Elchin Jafarov, Sofia Avendaño, and Bulbul Ahmmed provided reviews during the development of this article providing technical and editorial improvements.



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
