# Peer review of "New insights into the drainage of inundated Arctic polygonal tundra using fundamental hydrologic principles"

_The Cryosphere, 2020_

## Referee Comment (RC1) · Anonymous Referee #1 · 23 Jun 2020

I appreciated the opportunity to review this paper which broadened my perspective on polygonal tundra hydrology. It was particularly refreshing to read about analytical methods in contrast to the vast majority of studies relying on numerical models.

In the paper, the authors investigate a novel analytical model which conceptualizes the hydrological drainage dynamics of inundated ice-wedge polygon centres. The study is based on a related work by Zlotnik et al. (2020) that introduces the mathematics of the model, which is, however, not published yet. In the present article, the authors use the model to investigate how the pathways and the timing of drainage from polygon centres into polygon troughs depend on the ratio between polygon radius and thaw depth in the

centre, as well as on the ratio of hydraulic conductivities in the vertical and horizontal direction of the subsurface.

The paper addresses a relevant and timely topic, since thawing of permafrost which is expected to increase with Arctic climate warming, has substantial effects on polygonal tundra hydrology and nutrient cycling. The paper provides interesting insights into the hydrology of ice-wedge polygon centres, is written mostly in a concise and understandable way, and is certainly of interest to the audience of TC.

However, before publication of the paper, I see several points which deserve improvement. Addressing these points will hopefully improve the accessibility and the reach of the paper. I also found several smaller mistakes, particularly in the figures, which can probably be fixed easily. I hope that the authors find my comments useful for further improvement of this interesting study.

**General comments**

- The assumptions underlying the mathematical model as well as its applicability for real-world scenarios should be explained in more detail. Overall, the described setting seems to be highly idealized and does not capture key complexities such as other hydrological drivers like precipitation and evapotranspiration, or the dynamic evolution of thaw depths in the polygon rims, all of which have substantial influences on polygon drainage. It would be helpful if the assumptions regarding these complexities would be stated clearly (e.g., in the Model overview section). In this context, the paper would also benefit from mentioning specific real-world scenarios for its application (for example, drainage following a precipitation event, drainage over the entire thaw season, or changing drainage patters with a warming climate, etc.).

- The paper lacks a comprehensive discussion of the results. While the authors interpret their findings and mention potential implications for nutrient cycling, the
applicability and the limitations of the model deserve a more detailed discussion. In this context, it would for example be interesting to relate the findings to other mathematical (analytical and/or numerical) models addressing polygonal tundra hydrology (e.g., Cresto Aleina et al. (2013)). It would further be interesting to discuss in which way the approach could be transferred to other types of ice-wedge polygons, e.g. non-inundated low-centred polygons or high-centred polygons. To my opinion, a revised version of the paper would benefit from making use of subsections in the Discussion section, which is very hard to access in its present form.

- The Introduction section of the paper would benefit from making it more concise and clearly stating the research objectives of the paper towards its end. In the present version, the introduction appears to be "meandering" around a lot of distantly-related literature, without working out clearly the addressed research gap. To my opinion this section should be revised carefully, shortened where possible, and end with stating the objectives more clearly. It should also be clearly stated in which way the paper is different from or complementary to the paper of Zlotnik et al. (2020) which is still in review.

- In the present form, the paper's conclusions are presented as a repetition of the main results in a bullet-point style. In order to improve the overall accessibility of the article, the authors should complement their Conclusions by stating the context of their findings and providing take-home messages for the readers. In the present version, it is hard to understand the paper's conclusions in isolation from the rest of the study.

**Specific comments**

- The paper's title might be misleading as it refers to "drainage of inundated Arctic polygonal tundra", while the paper addresses drainage pathways of inundated

ice-wedge polygon centres, and not of the entire tundra landscape.

- I might have overlooked something, but I think that the assumption on the water level in the troughs (parameter $H_t$) is not stated in the paper. From the context I assume that a constant value is assumed for $H_t$, but this choice should be justified and the value should be given somewhere. This is also a potential point for the discussion, as a dynamically changing water level in the trough would presumable influence the drainage dynamics of the centres. In this context it could also be explained, why the water level cannot drop below the surface of the centre.

- To my opinion, the Methods section would be more intuitive if it was restructured such that it first provides an overview of the mathematics of the model, an then the assumptions, parameters values and the specific settings are stated.

- It might be helpful for readers to also state the full (dynamic) equation of the mathematical model in the Methods or in the appendix, which in the current version of the paper only contains the steady-state case.

- In lines 327 ff. the authors discuss the implications of drainage dynamics for the melting of the top of ice wedges. These explanations should be extended and put into relation to other aspects influencing the degradation of ice wedges (e.g., hydrologic regime of the troughs (Nitzbon et al. 2019), the geometry of the troughs (Abolt et al. 2020))

- Fig. 1: The inlet shows a polygon with hexagonal symmetry, while the mathematical model assumes a radial symmetry. This might confuse readers and the inlet should hence be adopted to reflect the mathematical model.

- The second panel in Fig. 2 and the second panel in Fig. 3 show the same cases. However, the reported numbers for the volume accessed by the flow

deviate slightly ($10.2\%$ vs. $10.6\%$). Only one number can be correct. The same applies to the reported numbers in the fourth panel of Fig. 3 and the second panel of Fig. 4 ($49.1\%$ vs. $43.5\%$). Please check which value is correct and report consistent numbers.

- In Fig. 5 and Fig. 8 it is hard to associated the colours displayed in the plot with those of the colourbar. The figure could be improved by using a continuous colour range in the plot and indicate isolines with numbers (similar to the labelled isolines in Fig. 2 to 4). It might also be worth considering to indicate the specific parameter combinations shown in Fig. 2,3,4,6,7 in the "global" Figures 5 and 8 with markers.

- The depletion times shown in the upper panel of Fig. 6 (case $K_r/K_z = 1$) do not match with the respective numbers for this case in the upper panel of Fig. 7. I suppose that the upper panel of Fig. 6 actually shows the case $K_r/K_z = 100$.

- The authors might consider combining Fig. 6 and 7 into one Figure or decide to not show Fig. 6 at all, as its information are also contained in Fig. 7. I also do not understand why the lines plotted in Fig. 7 do not use the same colour-coding as those shown in Fig. 6.

- If I understand correctly, the parameter $\kappa$ characterizes the hydraulic conductance of the rim, with higher values meaning an improved conductance of the rim. However, at some points (including Fig. 1) the parameter is referred to as the flow "resistance", which I find confusing (or I misunderstood the parameter). A consistent and intuitive terminology for this parameter should be used.

- The default value chosen for $\kappa$ should be mentioned and justified in the Methods section.

- The paper contains a reference to a Bachelor's thesis by Oehme (2019) which is wrongly stated as a Ph.D. thesis in the References list. The work by Oheme

(2019) is based on the numerical model by Nitzbon et al. (2019) published in TC, which might be an appropriate reference.

**Technical corrections**

- All panels of the Figures should be labelled with letters according to the journal standard (a,b,...), and these should be used in the main text for references to the Figures.

- Fig. 2: The precision of the decimal numbers on the left axes (e.g. $0.828L$) is higher than the respective values in Fig. 3, 4, etc. If there is no reason for this, this should be made consistent between the figure.

- The unit is missing in the y-axis label of Fig. 7.

- Units should be provided in the format required by TC.

- For the enumeration of boundary conditions in the text of Appendix A1, the authors should not use (1), (2), etc. as this might be confused with the numbering of the equations in the main text.

- Remove "TBD..." from the Author contributions statement. Contributions of EJ are not stated there either.

- Fig. 9: Should be $K_z$ and not $Kz$ in the label.

- Lines 224 and 225: Should be "upper right" and "lower right" instead of "ùpper left" and "upper right".

**References**

Cresto Aleina, F., Brovkin, V., Muster, S., Boike, J., Kutzbach, L., Sachs, T., Zuyev, S. (2013). A stochastic model for the polygonal tundra

based on Poisson–Voronoi diagrams. Earth Syst. Dynam., 4(2), 187–198. https://doi.org/10.5194/esd-4-187-2013

Abolt, C. J., Young, M. H., Atchley, A. L., Harp, D. R., Coon, E. T. (2020). Feedbacks Between Surface Deformation and Permafrost Degradation in Ice Wedge Polygons, Arctic Coastal Plain, Alaska. Journal of Geophysical Research: Earth Surface, 125(3), e2019JF005349. https://doi.org/10.1029/2019JF005349

---

## Referee Comment (RC2) · Anonymous Referee #2 · 24 Jun 2020

Low-centered ice-wedge polygon networks are nearly ubiquitous in Arctic lowlands, and thus a better understanding of ice-wedge polygon drainage patterns (timing and pathways) will inform on such issues as nutrient or carbon flushing from polygons, advective heat transport from polygon centres to the network of ice-wedge troughs that separate the polygons, and possibly on the transition from methane to carbon dioxide emissions with drainage and polygon drying. "New insights into the drainage of inundated Arctic polygonal tundra using fundamental hydrologic principles" by Harp et al. presents an investigation of inundated low-centered polygon drainage using a 3D-axisymmetric analytical model (Zlotnik et al. In Review). The authors purport this paper to be the first one to present a fundamental hydrological investigation of low-centered

ice-wedge polygon drainage. The authors importantly recognize that "fundamental hydrology dictates that ice-wedge polygon geometry and heterogeneity will explicitly govern subsurface drainage pathways and time spans", and explore the effects of polygon aspect ratio and vertical versus horizontal hydraulic conductivity anisotropy on hydrological dynamics of low-centered ice-wedge polygon drainage.

The model used is presented in another paper that is still under review, so as a reader I am not able to fully assess its merits, but there is a helpful abbreviated set of solutions in the Appendix. The model is based on the authors' assumption that an idealized polygon drainage domain is adequately represented by a cylinder, and that drainage will occur at a "vertical outer boundary" (P15, L255; Figure1). Drainage is allowed to occur uniformly in a ring around the cylinder as indicated by the maps shown of the percent of thawed soil accessed by 95% of the drainage flow (Figs. 2-4 and 9). The effect of aspect ratio on drainage is explored by altering the depth of the cylinder versus its radius. Recognizing that there are heterogeneities in internal hydraulic conductivities, the authors explore the effects of anisotropy by varying the ratio between vertical and horizontal hydraulic conductivities.

The paper is easy to read and results are clearly presented. It is actually quite a straightforward paper with effective figures. There are several minor corrections that I noticed, such as "MacKay" when it should be "Mackay", and the last sentence in the caption for Figure 4 is wrong, but nothing too distracting. I do find the title of the paper to be somewhat misleading as the focus of the work is on drainage of ponded water from within a single polygonal cell rather than drainage of polygonal tundra (the network of cells).

However, despite the generally good presentation, I think that there is a major problem that relates to the representation of the boundary conditions of the model's drainage domain. I began to wonder this as soon as I saw the diagram in Figure 1. The boundary that retains water within low-centered ice wedge polygons is bowl shaped, not cylindrical, and this bowl shape is constrained by the permafrost table that is mirrored in the

active layer by the frost table as it penetrates into the ground throughout the summer. If the boundary conditions of the model domain reflected a bowl-shaped geometry, there would likely be substantial implications on drainage, such as derived equipotential hydraulic heads or stream function, or the change in volume of ponded water within the polygon, as examples.

Water in a low-centered ice-wedge polygon is retained by the frozen core of the polygon rim, so flow occurs over the frozen rim of the bowl. Logically, the frozen rim of the bowl lowers over the thaw season as frost table progresses deeper into the ground, with drainage accompanying thaw-depth progression. Helbig et al. (2013), referenced by the present authors, indicate that the barrier function of polygon rims is strongly controlled by thaw. In the present paper, however, what each model run solves for is drainage potential of a cylinder full of water with a vertical outer boundary barrier determined by the given polygon thaw depth, L. That is, the model solutions are of "hydraulic heads and stream function in the thawed soil layer below a polygon center" (P6, L150). How can the cylinder retain water to depth L and develop the modelled hydraulic heads and streams if the ground has already thawed completely to depth L? Instantaneous thaw is not realistic. How does the system behave if the thaw occurs progressively downward at the vertical boundary? the model does not appear to be designed for transient boundary conditions, but perhaps transient boundary conditions could be represented by a set of stepwise models.

In any case, no matter how far into the thaw season the system is, water flow will always concentrate at any low spot in the rim of the bowl. This is critical. Wales et al. (2020), who are cited in the manuscript, show that drainage from low-centred ice-wedge polygons is very heterogeneous and preferential flow occurs at locations where the frost table is lowest. They state that "changing elevation of the frost table and its topography", though not shown in their conceptual diagram, "plays an important role in inhibiting infiltration and influencing preferential flow." Thus, the base case for low-centered ice-wedge polygon drainage is high heterogeneity in the polygon outer

boundary condition; however, Harp et al. consider such conditions to be "anomalous heterogeneities" (P4, L98). Instead they assume that homogeneous cylindrical geometry is the base case scenario. The clear demonstration that the focus of drainage at low points in the rim by Wales et al. (2020) breaks down the logic of the importance of aspect ratio in the cylindrical model as presented. Low-centered ice-wedge polygons have preferential flow locations at the rim, and are not dominated by diffuse flow across the entire rim, thus the cylindrical model as presented cannot stand as globally applicable. The overgeneralization of the model renders the results presented in Harp et al. to a limited state of usefulness.

Regarding the internal hydraulic conductivities, the authors recognize that radial and vertical hydraulic conductivities are often different, but it would be much more informative if the model domain were based more on reality and included, at minimum, a 2-layer system with an organic soil over a mineral soil. The authors describe this 2-layer system in the introduction (P. 3, L39), and it is typically the case (see for instance, Figure 1a, Wales et al. 2020). If a 2-layer system were replicated in the model, along with a more realistic boundary layer geometry, perhaps these changes may help explain the hydrology of low-centred ice-wedge polygons as observed in the field. Wales et al. (2020) find that the "general pattern in tracer dynamics in both polygon types was to first infiltrate vertically until it encountered the frost table, then to be transported horizontally, highlighting the influence of the frost table on horizontal flux." It is not clear how such field results can be represented according to the hydrological principals represented in the present model.

Finally, a comparatively minor point, even if there was no focus of flow at low points in the frost table and there was instantaneous thaw, I wondered how much the assumption of a circular planar geometry is really appropriate? An ice-wedge network is a set of polygons. Understandably, a circle is relatively straightforward to model, but most ice-wedge polygons are typically 4- to 6-sided. As the incircle radii for a unit area square, regular pentagon, regular hexagon, and circle are about 0.500, 0.525, 0.537,

and 0.564, respectively, I wondered if aspect ratios and resulting ponded water volumes modelled according to a cylinder should really be used to represent the planar geometry of polygons in the field? In other words, for a unit area, the perimeters of the polygons are greater than the circumference of a circle (~13% and ~ 8% greater for a square and regular pentagon, respectively), so flow across the outer boundary of the polygon becomes increasingly diffuse as the polygon order decreases. For example, would the estimates of ponded height depletion curves according to radius underestimate depletion rates with respect to natural geometries within polygonal tundra? There may not be that much difference with respect to drainage from a single polygon, but there probably is if one considers a network that is representative of Arctic polygonal tundra.

In summary, the focus of the paper is not really on the drainage of Arctic polygonal tundra, but on drainage of a single cell within Arctic polygonal tundra. The results presented in the manuscript are consistent with the model, but the model is overidealized and does not well represent base conditions of a single low-centred ice-wedge polygon. Therefore, the reader is left to conclude that the results and conclusions have limited implications. Until the above major points are addressed, I wouldn't recommend the paper for publication.

References

Helbig, M., Boike, J., Langer, M., Schreiber, P., Runkle, B. R., and Kutzbach, L.: Spatial and seasonal variability of polygonal tundra water balance: Lena River Delta, northern Siberia (Russia), Hydrogeology Journal, 21, 133–147, 2013.

Wales, N. A., Gomez-Velez, J. D., Newman, B. D., Wilson, C. J., Dafflon, B., Kneafsey, T. J., Soom, F., and Wullschleger, S. D.: Understanding the relative importance of vertical and horizontal flow in ice-wedge polygons, Hydrology and Earth System Sciences, 24, 1109–1129, 2020.

Zlotnik, V., Harp, D. R., and Abolt, C. J.: Edge effect on polygon drainage in permafrost

areas: implications for heat and mass transport, Water Resources Research, In Review.

---

## Author Comment (AC1) · 27 Jul 2020

Response to reviewers for tc-2020-100-RC1

We appreciate the thoughtful and helpful reviews that have allowed us to clarify and improve our manuscript. Some of the highlights of our improvements include:

- We made major changes to figures 5 and 8 using isolines and indicating locations on the response surfaces associated with other figures. This was a great suggestion from Reviewer 1 that pulls the figures of the paper together for the reader. We are very grateful for this suggestion.
- We added a plot to Figure 6 (now Figure 6b) that enriches the analysis of depletion curves with respect to high and low conductivity and provides a useful cross-reference to Figure 7a.
- We simplified and reorganized the Introduction to remove unnecessary material and more clearly present our research objectives.
- We now provide a more comprehensive discussion of the results with subsections in the Discussions section to improve reader accessibility.
- We have added context to the conclusions allowing them to be interpreted independent of the rest of the paper.
- We reorganized the Methods section to logically present the research.
- We now provide a clearer discussion of the objectives of our research.
- We now provide a complete discussion of model limitations and relation to other mathematical analyses.

Both reviewers recognized that the paper is "written mostly in a concise and understandable way", "easy to read and results are clearly presented", and "is actually quite a straightforward paper with effective figures". Therefore, with the additional improvements listed above as suggested by the reviewers, the manuscript is now accessible, clear, and concise. Please find our detailed responses to each comment below.

One point, mainly from Reviewer 2, focused on the limitations imposed by the simplifications of our model. We feel that the reviewer brings up a good point that deserves special attention here. We feel that comments related to the model limitations are mainly due to our previously not clearly stating our research objectives. We now clearly state our objectives and cast our research in its appropriate context on lines 3, 5, 14-16, 85, 97, 108-111, 114-134, and 422-448. We try to capture the gist of this discussion below.

Models are used for many purposes. These days, the focus is predominately on making predictions. However, models can also be used to test hypotheses and gain intuition about processes. The research in our manuscript fits into the latter category which is often overlooked these days. Our goal is not to demonstrate predictive prowess, but to use fundamental principles to gain intuition into the complex process of polygon drainage. By using a simplistic model, we are able to uncover insights into polygon drainage (e.g., the drainage edge effect) that would likely be obfuscated if we had used a complex model and tried to consider all the complexities of ice-wedge polygon geometry and hydraulic properties.

There are many complexities that will exist in ice-wedge polygons not considered in our manuscript that will alter the drainage patterns we present, but they will not reverse the fundamental insights drawn from idealized scenarios. We feel that future research investigating different ice-wedge polygon complexities with more complicated models will find this manuscript useful as a means to quantify and understand the deviations from idealized scenarios. In summary, we feel that using simplified models based on fundamental principles is a great way to gain understanding and intuition into complex processes that can otherwise seem unmanageable and incomprehensible. To make this point clearer to the reader, the manuscript has been recast to clearly state that our research objectives are to gain intuition and understanding and spur further research directions based on readers asking "what would the effect of this or that complexity be on these results?". We are thankful to the reviewers' comments that prompted this recasting, which is another significant improvement to the manuscript.

Note that references to line numbers below are for the manuscript with track changes.

Anonymous Referee #1

I appreciated the opportunity to review this paper which broadened my perspective on polygonal tundra hydrology. It was particularly refreshing to read about analytical methods in contrast to the vast majority of studies relying on numerical models.
In the paper, the authors investigate a novel analytical model which conceptualizes the hydrological drainage dynamics of inundated ice-wedge polygon centres. The study is based on a related work by Zlotnik et al. (2020) that introduces the mathematics of the model, which is, however, not published yet. In the present article, the authors use the model to investigate how the pathways and the timing of drainage from polygon centres into polygon troughs depend on the ratio between polygon radius and thaw depth in the centre, as well as on the ratio of hydraulic conductivities in the vertical and horizontal direction of the subsurface.
The paper addresses a relevant and timely topic, since thawing of permafrost which is expected to increase with Arctic climate warming, has substantial effects on polygonal tundra hydrology and nutrient cycling. The paper provides interesting insights into the hydrology of ice-wedge polygon centres, is written mostly in a concise and understandable way, and is certainly of interest to the audience of TC.

We are pleased that (1) our research broadened the reviewer's perspective, (2) that she/he recognizes the usefulness of analytical modeling to gain fundamental insights into Arctic hydrology, and (3) that she/he feels that the paper is certainly of interest to the TC audience!

However, before publication of the paper, I see several points which deserve improvement. Addressing these points will hopefully improve the accessibility and the reach of the paper. I also found several smaller mistakes, particularly in the figures, which can probably be fixed easily. I hope that the authors find my comments useful for further

improvement of this interesting study.

We thank the reviewer for a careful and thoughtful review of our manuscript. The review has resulted in several corrections and improvements to figures and recasting of text to improve reader accessibility. Please see below for our individual responses to comments.

General comments
• The assumptions underlying the mathematical model as well as its applicability for real-world scenarios should be explained in more detail. Overall, the described setting seems to be highly idealized and does not capture key complexities such as other hydrological drivers like precipitation and evapotranspiration, or the dynamic evolution of thaw depths in the polygon rims, all of which have substantial influences on polygon drainage. It would be helpful if the assumptions regarding these complexities would be stated clearly (e.g., in the Model overview section). In this context, the paper would also benefit from mentioning specific real-world scenarios for its application (for example, drainage following a precipitation event, drainage over the entire thaw season, or changing drainage patters with a warming climate, etc.).

We now provide text in the introduction and discussions sections clearly stating that the objective or our analysis is to expose fundamental hydrologic insights into inundated polygon drainage, not provide a predictive capability that can be applied across polygon scenarios. Our use of non-dimensionalized heads and ponded heights facilitate this type of analysis, providing a powerful approach to gain fundamental understanding without becoming overwhelmed with individual scenario specifics. This allows the reader to use the idealized conceptualizations of ice-wedge polygons to gain intuition about drainage patterns and timing. In Section 4.2 "Model limitations" section (lines 422-448), we discuss in what scenarios and during what timeframes the model can be applied. In the introduction, we now explain that while we do not consider evaporation and precipitation, these will not alter the drainage patterns nor a relative comparison of drainage times (based on having identical precipitation and evaporation across scenarios) (lines 121-129). We find this simplification perfectly reasonable for the purposes of gaining intuition on how polygons drain in our sensitivity analysis. We now explain the implications of additional parameters including the kappa term and H_t within our model (lines 210-248).

• The paper lacks a comprehensive discussion of the results. While the authors interpret their findings and mention potential implications for nutrient cycling, the applicability and the limitations of the model deserve a more detailed discussion. In this context, it would for example be interesting to relate the findings to other mathematical (analytical and/or numerical) models addressing polygonal tundra hydrology (e.g., Cresto Aleina et al. (2013)). It would further be interesting to discuss in which way the approach could be transferred to other types of ice-wedge polygons, e.g. non-inundated low-centred polygons or high-centred polygons. To my opinion, a revised version of the paper would benefit from making use of subsections in the Discussion section, which is very hard to access in its present

form.

We have added subsections to the Discussion section as follows:

4.1 Analysis implications

4.2 Model limitations

4.3 Relation to other mathematical analyses

We agree with the reviewer that this makes it easier for readers to access.

In Section 4.2, we now discuss the applicability of the model to other polygon scenarios (lines 445-448):

"Since the model is based on the saturated groundwater flow equation, in its current form it cannot be applied to non-inundated low-centered polygons. Since it is based on having a ponded center, the model is also not applicable to high-centered polygons. However, a similar approach to that presented here could be taken with the unsaturated groundwater flow equation to capture these other polygon scenarios and types."

We have added comparisons to other mathematical approaches including Cresto Aleina in Section 4.3. This research is also mentioned in the Introduction section now (lines 74-79).

• The Introduction section of the paper would benefit from making it more concise and clearly stating the research objectives of the paper towards its end. In the present version, the introduction appears to be "meandering" around a lot of distantly-related literature, without working out clearly the addressed research gap. To my opinion this section should be revised carefully, shortened where possible, and end with stating the objectives more clearly. It should also be clearly stated in which way the paper is different from or complementary to the paper of Zlotnik et al. (2020) which is still in review.

We have shortened and restructured the Introduction based on the reviewer's comments. Unnecessary material has been removed (lines 35-37, 41-43, 65-69, and 112-128) and structure added. The revised structure of the Introduction is:

Paragraph 1: Describes the problem and why the reader should care.

Paragraph 2: Describes the polygonal tundra landscape, how it developed, and how that governs its present-day hydrology.

Paragraph 3 and 4: Review the Arctic hydrology community's current understanding and conceptualization of inundated low-centered polygon hydrology.

Paragraph 5: Describes what is missing from the currently available research.

Paragraph 6 and 7: Describe our research and explain how it helps address certain gaps in the currently available research.

The Introduction is now concise and the objectives of our research are clearly stated and organized. We thank the reviewer for prompting a reanalysis of our Introduction.

• In the present form, the paper's conclusions are presented as a repetition of the main results in a bullet-point style. In order to improve the overall accessibility of the article, the authors should complement their Conclusions by stating the context of their findings and providing take-home messages for the readers. In

the present version, it is hard to understand the paper's conclusions in isolation from the rest of the study.

We added a paragraph to set the context for the conclusions and added contextual take-home messages to each conclusion. We have left the bullet-point style since we assume that this was not the main issue for the reviewer, and feel that this format helps organize the conclusions into take-home messages. We thank the reviewer for indicating that the conclusions could be improved.

Specific comments
• The paper's title might be misleading as it refers to "drainage of inundated Arctic polygonal tundra", while the paper addresses drainage pathways of inundated ice-wedge polygon centres, and not of the entire tundra landscape.

We have changed the title to "New insights into the drainage of inundated ice-wedge polygons using fundamental hydrologic principles". We completely agree that the old title was misleading and that the new title is more appropriate. We thank the reviewer for pointing this out.

• I might have overlooked something, but I think that the assumption on the water level in the troughs (parameter Ht) is not stated in the paper. From the context I assume that a constant value is assumed for Ht, but this choice should be justified and the value should be given somewhere. This is also a potential point for the discussion, as a dynamically changing water level in the trough would presumable influence the drainage dynamics of the centres. In this context it could also be explained, why the water level cannot drop below the surface of the centre.

We agree that we did fail to mention the value of Ht. We now justify and state on lines 226-248 that its value has been set equal to the polygon center ground surface in all cases. We thank the reviewer for bringing this to our attention. We now also explain how Ht can be above or below the ground surface and the effect this has on the solution:

"In practice, the water level in troughs (Ht in equations 1 and 3) will vary throughout the thaw season, effecting the magnitude of heads in the soil of the polygon center and drainage times. As the non-dimensional heads are relative to Ht (refer to equation A7), its value does not affect our comparisons of drainage patterns (which are based on non-dimensional heads that are normalized from 0 to 1). The value of Ht will affect our comparisons of drainage time, but in a systematic, interpretable manner. For example, a higher Ht will compress the exponential curve defined by equation 3 upwards, while a lower Ht will extend the exponential curve downwards. In cases where Ht is below the polygon center ground surface, the solution is valid until Hc reaches the ground surface, at which time the ponded center has completely drained. Therefore, to isolate our analysis to aspect ratio and anisotropy, we have set Ht in all cases to the ground surface of the polygon center."

• To my opinion, the Methods section would be more intuitive if it was restructured such that it first provides an overview of the mathematics of the model, an then

the assumptions, parameters values and the specific settings are stated.
We agree and have switched the ordering of the subsections in the Methods section, and now start with the "Model overview" followed by the "Model parameterization" subsection.

• It might be helpful for readers to also state the full (dynamic) equation of the mathematical model in the Methods or in the appendix, which in the current version of the paper only contains the steady-state case.
The full transient equation for hydraulic heads is now presented as equation A1.

• In lines 327 ff. the authors discuss the implications of drainage dynamics for the melting of the top of ice wedges. These explanations should be extended and put into relation to other aspects influencing the degradation of ice wedges (e.g., hydrologic regime of the troughs (Nitzbon et al. 2019), the geometry of the troughs (Abolt et al. 2020))
We agree with the reviewer that our findings on potential advective heat transport towards ice wedge tops are of interest within the context of other recent research. We have added a sentence on lines 411-413 indicating the other research perspectives mentioned above by the reviewer on ice wedge degradation. We do not provide a detailed discussion drawing links between these papers since our research looks at inundated polygons, so the hydrologic regime will be wet in all of our cases, and we don't consider trough geometry. This makes drawing links between our findings and these papers inappropriate, although we do agree that they provide multiple perspectives to consider when trying to understand ice-wedge degradation.

• Fig. 1: The inlet shows a polygon with hexagonal symmetry, while the mathematical model assumes a radial symmetry. This might confuse readers and the inlet should hence be adopted to reflect the mathematical model.
We have removed the inset to avoid this confusion.

• The second panel in Fig. 2 and the second panel in Fig. 3 show the same cases. However, the reported numbers for the volume accessed by the flow deviate slightly (10.2% vs. 10.6%). Only one number can be correct. The same applies to the reported numbers in the fourth panel of Fig. 3 and the second panel of Fig. 4 (49.1% vs. 43.5%). Please check which value is correct and report consistent numbers.
We thank the reviewer for catching this. The annotations have been corrected and are now consistent between Fig2b and Fig3b and between Fig3d and Fig4b.

• In Fig. 5 and Fig. 8 it is hard to associated the colours displayed in the plot with those of the colourbar. The figure could be improved by using a continuous colour range in the plot and indicate isolines with numbers (similar to the labelled isolines in Fig. 2 to 4). It might also be worth considering to indicate the specific parameter combinations shown in Fig. 2,3,4,6,7 in the "global" Figures 5 and 8 with markers.

These are great suggestions! We have modified the figures as suggested by the reviewer. This significantly enhances the manuscript and ties together nearly all of the figures! Thank You! The annotated points indicating Figure subplots are now referenced in the text on line 290.

• The depletion times shown in the upper panel of Fig. 6 (case Kr/Kz = 1) do not match with the respective numbers for this case in the upper panel of Fig. 7. I suppose that the upper panel of Fig. 6 actually shows the case Kr/Kz = 100.
The reviewer is exactly correct. The values in Fig. 7a (Kr/Kz=1 line) do not match those in Fig. 6a. This is because, while the anisotropy is equivalent, the magnitude of Kr and Kz are not. To make this clear to the reader, we have added a plot to Fig 6 (now Fig. 6b) where the magnitudes of the conductivities are equal to those in Fig. 7a. Now the reader can cross reference the Kr/Kz=1 line in Fig. 7a with Fig. 6b and see that they do indeed match. This not only provides consistency between Figures 6 and 7, but also brings a more complete analysis to Figure 6 presenting the effect of hydraulic conductivity magnitude on drainage time. We are extremely grateful for the careful attention of the reviewer in catching this discrepancy and prompting these clarifying improvements to the manuscript.

• The authors might consider combining Fig. 6 and 7 into one Figure or decide to not show Fig. 6 at all, as its information are also contained in Fig. 7. I also do not understand why the lines plotted in Fig. 7 do not use the same colour-coding as those shown in Fig. 6.
The information in Fig. 7 is intended to illustrate that while aspect ratio has a nearly linear effect on drainage time, the effect due to anisotropy is exponential. A reader might deduce this from Fig. 6 by inspecting the characteristic times indicated along the x axis, but not likely. We therefore feel that Fig. 7 does add to the information that a reader acquires, but that we need to make the point clearer. To do this, we have color coded the lines as the reviewer suggests and now indicate in the Figure 7 caption the linear vs exponential dependence of drainage time to aspect ratio and anisotropy, respectively.

• If I understand correctly, the parameter $\kappa$ characterizes the hydraulic conductance of the rim, with higher values meaning an improved conductance of the rim. However, at some points (including Fig. 1) the parameter is referred to as the flow "resistance", which I find confusing (or I misunderstood the parameter). A consistent and intuitive terminology for this parameter should be used.
We can see how our terminology that $\kappa$ "characterizes the resistance" could be confusing. We have changed all instances to "conductance" now (line 153, 154, 155, 220, 221, 333, 352, 527, 528, and Figure 9 caption).

• The default value chosen for $\kappa$ should be mentioned and justified in the Methods section.
The default value of 5 s$^{-1}$ is now stated and justified on lines 220-223.

• The paper contains a reference to a Bachelor's thesis by Oehme (2019) which is wrongly stated as a Ph.D. thesis in the References list. The work by Oheme

(2019) is based on the numerical model by Nitzbon et al. (2019) published in TC, which might be an appropriate reference.

The reference has been changed to Nitzbon et al. (2019) (line 59).

Technical corrections

• All panels of the Figures should be labelled with letters according to the journal standard (a,b,...), and these should be used in the main text for references to the Figures.

All figures with multiple plots now have the subplots alphabetically labelled and these are now used throughout the text.

• Fig. 2: The precision of the decimal numbers on the left axes (e.g. 0.828L) is higher than the respective values in Fig. 3, 4, etc. If there is no reason for this, this should be made consistent between the figure.

The precision of the annotation is now consistent across all figures. The reason we decided to have more precision in the left axes of Fig. 2 was because the values in (b) and (c) are same at 2 decimal places, but different at 3. However, in retrospect, the difference in the third decimal place is not important for the purposes of the figure, and the reader probably infers that they are different beyond the precision presented. While we were at it, we reduced the precision of other annotation in all figures to 2 significant digits (note that in some cases, 1 or 3 significant digits are used to maintain consistent precision across subplots when magnitudes differ). We feel this removes clutter from the figures without loss of any critical information. We thank the reviewer for spurring this second look at the precision in our figure annotation.

• The unit is missing in the y-axis label of Fig. 7.

This has been added.

• Units should be provided in the format required by TC.

All units now use the TC format.

• For the enumeration of boundary conditions in the text of Appendix A1, the authors should not use (1), (2), etc. as this might be confused with the numbering of the equations in the main text.

We now use BC1, BC2,…,BC5 to avoid this confusion.

• Remove "TBD..." from the Author contributions statement. Contributions of EJ are not stated there either.

The Author contribution statement is now completed.

• Fig. 9: Should be $K_z$ and not Kz in the label.

This has been corrected.

• Lines 224 and 225: Should be "upper right" and "lower right" instead of "ùpper left" and "upper right".

This has been corrected. We thank the reviewer for catching this.

References
Cresto Aleina, F., Brovkin, V., Muster, S., Boike, J., Kutzbach, L., Sachs, T., Zuyev, S. (2013). A stochastic model for the polygonal tundra based on Poisson–Voronoi diagrams. Earth Syst. Dynam., 4(2), 187–198. https://doi.org/10.5194/esd-4-187-2013
Abolt, C. J., Young, M. H., Atchley, A. L., Harp, D. R., Coon, E. T. (2020). Feedbacks Between Surface Deformation and Permafrost Degradation in Ice Wedge Polygons, Arctic Coastal Plain, Alaska. Journal of Geophysical Research: Earth Surface, 125(3), e2019JF005349. https://doi.org/10.1029/2019JF005349
Anonymous Referee #2

Low-centered ice-wedge polygon networks are nearly ubiquitous in Arctic lowlands, and thus a better understanding of ice-wedge polygon drainage patterns (timing and pathways) will inform on such issues as nutrient or carbon flushing from polygons, advective heat transport from polygon centres to the network of ice-wedge troughs that separate the polygons, and possibly on the transition from methane to carbon dioxide emissions with drainage and polygon drying. "New insights into the drainage of inundated Arctic polygonal tundra using fundamental hydrologic principles" by Harp et al. presents an investigation of inundated low-centered polygon drainage using a 3Daxisymmetric analytical model (Zlotnik et al. In Review). The authors purport this paper to be the first one to present a fundamental hydrological investigation of low-centered ice-wedge polygon drainage. The authors importantly recognize that "fundamental hydrology dictates that ice-wedge polygon geometry and heterogeneity will explicitly govern subsurface drainage pathways and time spans", and explore the effects of polygon aspect ratio and vertical versus horizontal hydraulic conductivity anisotropy on hydrological dynamics of low-centered ice-wedge polygon drainage.

We appreciate that the reviewer recognizes the importance of using "fundamental hydrology" to understand polygon drainage. We agree that starting out with the fundamentals can provide clarity and basic understanding, which then provides direction for investigating further complexities. This is an easy point to lose sight of with hydrology's current push to use complex models, where researchers often feel compelled to start out with a high degree of complexity. Unfortunately, starting out with complex models often results in obfuscation of the fundamental drivers and characteristics of the problem which can be revealed by starting out by investigating the fundamentals of the problem. This is an important point to keep in mind when evaluating the remainder of this review and our responses.

The model used is presented in another paper that is still under review, so as a reader I am not able to fully assess its merits, but there is a helpful abbreviated set of solutions

in the Appendix. The model is based on the authors' assumption that an idealized polygon drainage domain is adequately represented by a cylinder, and that drainage will occur at a "vertical outer boundary" (P15, L255; Figure1). Drainage is allowed to occur uniformly in a ring around the cylinder as indicated by the maps shown of the percent of thawed soil accessed by 95% of the drainage flow (Figs. 2-4 and 9). The effect of aspect ratio on drainage is explored by altering the depth of the cylinder versus its radius. Recognizing that there are heterogeneities in internal hydraulic conductivities, the authors explore the effects of anisotropy by varying the ratio between vertical and horizontal hydraulic conductivities.

The reviewer's summary of our approach is accurate. We would just add that the "vertical outer boundary" is a Robin boundary, allowing both the head and flux to be specified, which allows resistance to drainage (i.e., a skin effect) across the boundary. This is important in representing the resistance to drainage caused by elevated frozen ground often present under rims and the accumulation of fines along the soil/water interface of the trough. This is an important addition to make to the reviewer's summary that will enter into some of the discussion below.

The paper is easy to read and results are clearly presented. It is actually quite a straightforward paper with effective figures. There are several minor corrections that I noticed, such as "MacKay" when it should be "Mackay", and the last sentence in the caption for Figure 4 is wrong, but nothing too distracting. I do find the title of the paper to be somewhat misleading as the focus of the work is on drainage of ponded water from within a single polygonal cell rather than drainage of polygonal tundra (the network of cells).

We thank the reviewer for recognizing that our paper "is easy to read and results are clearly presented" and that it is a "straightforward paper with effective figures". This is in large part a result of using fundamental analyses with a minimum of complexities allowing a clear presentation. Of course, complexities need to ultimately be considered, but the fact that they would obfuscate the insights gleaned from our analysis at this stage should be kept in mind in the discussions below.

"MacKay" has been corrected to "Mackay".
The caption for Figure 4 has been fixed, we thank the reviewer for indicating this.
The title has been changed to "New insights into the drainage of *inundated ice-wedge polygons* using fundamental hydrologic principles", which we agree is more appropriate.

However, despite the generally good presentation, I think that there is a major problem that relates to the representation of the boundary conditions of the model's drainage domain. I began to wonder this as soon as I saw the diagram in Figure 1. The boundary that retains water within low-centered ice wedge polygons is bowl shaped, not cylindrical, and this bowl shape is constrained by the permafrost table that is mirrored in the active layer by the frost table as it penetrates into the ground throughout the summer. If the boundary conditions of the model domain reflected a bowl-shaped geometry, there would likely be substantial implications on drainage, such as derived equipotential hydraulic heads or stream function, or the change in volume of ponded water within the

polygon, as examples.

Yes, we agree that the thaw table in low-centered polygons will typically have raised rims that are a muted reflection of the surface topography above. Since the frost table in general follows the surface topography, and is often dampened compared to the surface, many low-centered polygons have relatively flat frost tables until close to the rims. In fact, this has been observed near Utqiagvik, AK (Romanovsky et al. (2017); Cable (2016)). Therefore, "bowl shaped" may imply more curvature then typically exists, perhaps "dish shaped" would be more evocative of the typical shape? To account for the raised frost table under the rims (which are outside our model domain), our model uses a Robin boundary condition that allows for a different effective conductance below the rim than in the center. This allows the model to represent a decreased conductance under the rim due to the flow constriction caused by raised frozen ground and other factors. We now explain this in detail on lines 93-96, 152-157, 210-223, and 330-335.

Romanovsky, V., Cable, W., and Dolgikh, K.: Subsurface Temperature, Moisture, Thermal Conductivity and Heat Flux, Barrow, Area A, B, C, D. Next Generation Ecosystem Experiments Arctic Data Collection, Oak Ridge National Laboratory, US Department of Energy, https://doi.org/10.5440/1126515, 2017.

Cable, W. L.: The role of environmental factors in regional and local scale variability in permafrost thermal regime, Masters thesis, University of Alaska Fairbanks, USA, 86 pp., 2016.

Water in a low-centered ice-wedge polygon is retained by the frozen core of the polygon rim, so flow occurs over the frozen rim of the bowl. Logically, the frozen rim of the bowl lowers over the thaw season as frost table progresses deeper into the ground, with drainage accompanying thaw-depth progression. Helbig et al. (2013), referenced by the present authors, indicate that the barrier function of polygon rims is strongly controlled by thaw. In the present paper, however, what each model run solves for is drainage potential of a cylinder full of water with a vertical outer boundary barrier determined by the given polygon thaw depth, L. That is, the model solutions are of "hydraulic heads and stream function in the thawed soil layer below a polygon center" (P6, L150). How can the cylinder retain water to depth L and develop the modelled hydraulic heads and streams if the ground has already thawed completely to depth L? Instantaneous thaw is not realistic. How does the system behave if the thaw occurs progressively downward at the vertical boundary? the model does not appear to be designed for transient boundary conditions, but perhaps transient boundary conditions could be represented by a set of stepwise models.

Yes, the reviewer's last statement is exactly correct. Insights into the transient nature of drainage can be represented by stepwise models, particularly given the relatively small temporal changes compared to spatial changes in heads. This is precisely the point of Figures 2, 4, and horizontal transects through Figure 5. In these figures, a decreasing aspect ratio is analogous to stepwise models of thaw depth increases. This is indicated on the x-axis label and in the caption of Figure 5. This is also stated and explained on lines 98-101, 186-188, 269-271, and 377-379. With regard to the reviewer's question as to how "can the cylinder retain water to depth L and develop the modelled hydraulic heads and streams if the ground has already

thawed completely to depth L?", we are not assuming "instantaneous thaw". Instead, we are showing scenarios which are known to exist in low-centered polygons: ponded conditions at different times throughout the thaw season at different thaw depths. As we present non-dimensional heads, discussed on lines 138-141, the drainage patterns apply to an arbitrary ponded height. Therefore, the results can be interpreted without the need to invoke an assumption of "instantaneous thaw".

In any case, no matter how far into the thaw season the system is, water flow will always concentrate at any low spot in the rim of the bowl. This is critical. Wales et al. (2020), who are cited in the manuscript, show that drainage from low-centred ice-wedge polygons is very heterogeneous and preferential flow occurs at locations where the frost table is lowest. They state that "changing elevation of the frost table and its topography", though not shown in their conceptual diagram, "plays an important role in inhibiting infiltration and influencing preferential flow." Thus, the base case for low-centered ice-wedge polygon drainage is high heterogeneity in the polygon outer boundary condition; however, Harp et al. consider such conditions to be "anomalous heterogeneities" (P4, L98). Instead they assume that homogeneous cylindrical geometry is the base case scenario. The clear demonstration that the focus of drainage at low points in the rim by Wales et al. (2020) breaks down the logic of the importance of aspect ratio in the cylindrical model as presented. Low-centered ice-wedge polygons have preferential flow locations at the rim, and are not dominated by diffuse flow across the entire rim, thus the cylindrical model as presented cannot stand as globally applicable. The overgeneralization of the model renders the results presented in Harp et al. to a limited state of usefulness.

Yes, we agree that low points along the frost table beneath the rims will lead to preferential flow paths. We refer to these, and the many other varied heterogeneities that will exist in ice-wedge polygons, as "anomalous heterogeneities" to indicate that they are at a scale beyond which can be captured by effective properties. We did not intend to imply that they are low probability scenarios. To eliminate this confusion, we have removed the term "anomalous " in all cases. We agree, in reality, things will be much more complicated than our idealized scenarios. A low spot in the frost table under the rim will obviously warp our drainage patterns in 3D space in ways that our model cannot represent in exact details. However, it will not reverse the fundamental intuition that is gained from looking at idealized scenarios. For example, we suspect that an edge effect will still be present along with many of the other characteristics that our solutions present. In some cases, the low spot in the rim frost table will be subtle, and our drainage patterns will be generally representative. In other cases where, for some reason, an extreme low spot in the rim frost table has developed, deviations from our drainage patterns will be greatest. If the point is to gain intuition, understanding idealized scenarios is extremely useful as they not only give you intuition into fundamentals, they also spur questions about how results will change under non-idealized scenarios. We therefore agree that our results need to be viewed with the understanding that complexities such as low spots in the frost table rims will exist, but within that context, the results provide a great deal of information for researchers. To this effect, our objective, the limitations of the model, and utility of our results are now clearly stated on lines 108-133 and 422-448.

It is also important to note that the surface of the frost table during Wales' tracer experiment is not known. Wales et al. (2020) are postulating about the frost table topography as a possible explanation of the breakthrough curves they observed.
The last argument in favor of the model is well documented erosion of the rims. In the case of focused discharge/discharge through the localized high-K heterogeneity, the erosion would be present in localized way near this heterogeneity and remain there over the years. However geomorphological data show that erosion is reasonably uniform over the circumference of the polygon (Abolt et al., 2017).

Abolt, C.J., Young, M.H., and Caldwell, T.G.: Numerical modelling of ice-wedge polygon geomorphic transition, Permafrost and Periglacial Processes, 28, 347–355, 2017 (see their Figure 4)

Regarding the internal hydraulic conductivities, the authors recognize that radial and vertical hydraulic conductivities are often different, but it would be much more informative if the model domain were based more on reality and included, at minimum, a 2-layer system with an organic soil over a mineral soil. The authors describe this 2-layer system in the introduction (P. 3, L39), and it is typically the case (see for instance, Figure 1a, Wales et al. 2020). If a 2-layer system were replicated in the model, along with a more realistic boundary layer geometry, perhaps these changes may help explain the hydrology of low-centred ice-wedge polygons as observed in the field. Wales et al. (2020) find that the "general pattern in tracer dynamics in both polygon types was to first infiltrate vertically until it encountered the frost table, then to be transported horizontally, highlighting the influence of the frost table on horizontal flux." It is not clear how such field results can be represented according to the hydrological principals represented in the present model.
Yes, representing a two (or more) layer system would be an interesting extension of our research, However, this would not change the structure of streamline kinematics if these layers are uniform (high-K above low-K or vise versa) as known from the early work of Freeze and Witherspoon in the late 60s. As far as the results from Wales et al. (2020), the polygon in their tracer experiment has some significant deviations from our model. First and foremost, the polygon was not inundated when the tracer was applied. Then, after application of the tracer, water was applied to help flush the tracer into the subsurface. So, there was unsaturated conditions followed by an unsaturated infiltration event. Additionally, vertical movement of the tracer could be attributed to the density effect, when tracer was injected in high concentrations (on the order of 1000 mg/L or above) resulting in a free convection phenomenon at the early stage and horizontal movement at later stages of the tracer test. They were also dealing with a limited number of sensors limiting their ability to map out tracers throughout the polygon. These factors need to be considered when comparing the results of Wales et al. and our drainage patterns. So, as the reviewer questions, it is not possible, appropriate, or conclusive to evaluate the results of the Wales et al. (2020) tracer experiment with our model. Here again, representing multi-layered systems (not to mention mixed, cryoturbated soils) is an added complexity to be considered. Of course, a thorough analysis of a layered soil profile would require the analysis of many scenarios. Large and small disparities between properties, etc. All

of these complexities are relevant and interesting, and are among the myriad of scenarios that our results bring to mind and make you wonder how they will affect the results. We agree that all of these complexities should be investigated, but don't feel that this introductory manuscript could provide the comprehensive treatment these complexities warrant. The context of the paper as a means to gain fundamental intuition, as opposed to a complete treatment of all possible complexities, is now described on lines 3, 5, 14-16, 85, 97, 108-111, 114-134, and 422-448.

Freeze RA, Witherspoon PA (1967) Theoretical analysis of regional groundwater flow. 2. Effect of water table configuration and subsurface permeability variation. Water Resources Research 3(2):623-634.

Finally, a comparatively minor point, even if there was no focus of flow at low points in the frost table and there was instantaneous thaw, I wondered how much the assumption of a circular planar geometry is really appropriate? An ice-wedge network is a set of polygons. Understandably, a circle is relatively straightforward to model, but most ice-wedge polygons are typically 4- to 6-sided. As the incircle radii for a unit area square, regular pentagon, regular hexagon, and circle are about 0.500, 0.525, 0.537, and 0.564, respectively, I wondered if aspect ratios and resulting ponded water volumes modelled according to a cylinder should really be used to represent the planar geometry of polygons in the field? In other words, for a unit area, the perimeters of the polygons are greater than the circumference of a circle (_13% and _ 8% greater for a square and regular pentagon, respectively), so flow across the outer boundary of the polygon becomes increasingly diffuse as the polygon order decreases. For example, would the estimates of ponded height depletion curves according to radius underestimate depletion rates with respect to natural geometries within polygonal tundra? There may not be that much difference with respect to drainage from a single polygon, but there probably is if one considers a network that is representative of Arctic polygonal tundra.

We agree that this is an interesting point. Of course, we don't claim predictive prowess of our model in all possible situations, and instead are looking at relative changes between scenarios in a sensitivity analysis. However, conceptual analysis of rectangular polygons indicate the same physical foundations of the edge effect. Mathematically such an analysis would be more cumbersome, and include unnecessary details of full 3D calculations that would be difficult to visualize lucidly. We agree that this point should be considered when evaluating our depletion curves. We mention this in section 4.2 "Model limitations" on lines 422-448.

In summary, the focus of the paper is not really on the drainage of Arctic polygonal tundra, but on drainage of a single cell within Arctic polygonal tundra. The results presented in the manuscript are consistent with the model, but the model is overidealized and does not well represent base conditions of a single low-centred ice-wedge polygon. Therefore, the reader is left to conclude that the results and conclusions have limited implications. Until the above major points are addressed, I wouldn't recommend the paper for publication.

We have changed the title to better represent the focus of the paper on a single inundated ice-wedge polygon. We also argue that the idealized scenarios investigated in the manuscript allow for fundamental insights to be drawn concerning the drainage of ice-wedge polygons. Insights such as the edge effect of polygon drainage have been missed by many researchers (including many of the co-authors of this manuscript) using more complicated models that can consider much more complexity, but obfuscate fundamental hydrology. Therefore, while we agree that the results are of "limited application" to specific individual scenarios, they are of wide application to gaining intuition and spuring further questions couched in the knowledge of the fundamental hydrology of idealized polygon scenarios. The context of the paper as a means to gain fundamental intuition, as opposed to a complete treatment of all possible complexities, is now described on lines 3, 5, 14-16, 85, 97, 108-111, 114-134, and 422-448.

References
Helbig, M., Boike, J., Langer, M., Schreiber, P., Runkle, B. R., and Kutzbach, L.: Spatial and seasonal variability of polygonal tundra water balance: Lena River Delta, northern Siberia (Russia), Hydrogeology Journal, 21, 133–147, 2013.
Wales, N. A., Gomez-Velez, J. D., Newman, B. D., Wilson, C. J., Dafflon, B., Kneafsey, T. J., Soom, F., and Wullschleger, S. D.: Understanding the relative importance of vertical and horizontal flow in ice-wedge polygons, Hydrology and Earth System Sciences, 24, 1109–1129, 2020.
Zlotnik, V., Harp, D. R., and Abolt, C. J.: Edge effect on polygon drainage in permafrost areas: implications for heat and mass transport, Water Resources Research, In Review.